# Towards Active Synthetic Data Generation for Finetuning Language Models

## Abstract

A common and effective means for improving language model capabilities involves finetuning a "student" language model's parameters on generations from a more proficient "teacher" model. Termed "synthetic data", these generations are often produced before any student finetuning, but some work has considered generating new synthetic samples as training progresses. This paper studies and advocates for the latter case, where data are generated in an iterative, closed-loop fashion that is guided by the current state of the student model. For a fixed budget of generated samples, or a budget in terms of compute spent querying a teacher, we show that this curation of finetuning data affords improved student performance over static generation. Further, while there have been several LLM-specific methods proposed that operate in this regime, we find that simple, inexpensive selection criteria from the active learning literature tend to be most performant. We validate these claims across four mathematical and logical reasoning datasets using four different small language models.

## 1 Introduction

Large Language Models (LLMs) have shown remarkable abilities in a wide variety of reasoning and factual knowledge tasks (Achiam et al., 2023; Bubeck et al., 2023; Katz et al., 2024), but their large size makes inference expensive. With the advent of agentic systems that interact with the external world, LLMs are poised to become even more ubiquitous in science, technology, and society, but the tremendous inference cost presents a challenge for realizing the full potential of these agents.

One way to quell the computational expense associated with LLM inference is to use small language models (SLMs). With orders of magnitude fewer parameters, SLMs are faster, cheaper, and easier to finetune for specialised skills like tool use, making them natural specialists using proprietary data or within agentic systems (Belcak et al., 2025).

Training language models typically involves three stages: pre-training on large general-purpose corpora, supervised finetuning (SFT), and reinforcement learning from human feedback (RLHF) or from verifiable rewards (RLVR) (Ouyang et al., 2022). SFT, the focus of this work, is critical for adapting a base model to a target distribution, and is especially common when training SLMs to improve their task-specific performance.

However, real-world data for SFT can be hard to obtain, or may lack desirable properties such as chain-of-thought reasoning (Wei et al., 2022). Consequently, a typical strategy involves synthesizing a corpus of question and answer pairs from a larger, more capable model (Mitra et al., 2024; Liu et al., 2024a). This process usu-

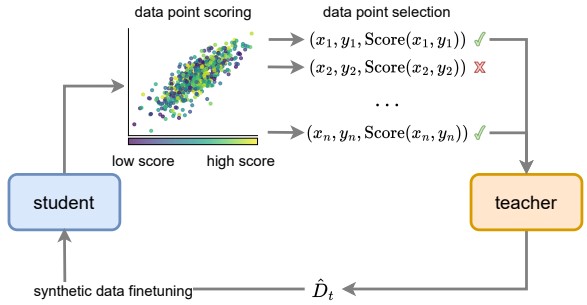

Figure 1: **Overview of iterative synthetic data generation (Algorithm 1)**. The student model guides synthetic data generation by prioritizing which data are used as an example for the teacher model to generate a new synthetic data point (Section 4.2). The student finetunes on teacher generated synthetic data.

ally begins with a small seed dataset and leverages a teacher LLM to produce supplementary synthetic samples, then finetunes the student SLM on the resulting sequences in aggregate.

Still, evidence suggests that generating a large static synthetic dataset is frequently wasteful, as it can often be drastically pruned with little to no degradation in trained model capabilities (Chen et al., 2023; Zhou et al., 2024). As such, this paper explores an iterative, targeted approach to synthetic data generation that is student-aware and improves data efficiency—achieving stronger performance under a fixed data generation budget than naive, static generation—thereby yielding a superior performance–training-set-size Pareto frontier (see Section 2 for a formal definition).

To facilitate productive learning, this work studies how we can effectively cater to the state of the student model and guide synthetic data generation by a teacher LLM via prompting (Mitra et al., 2024; Liu et al., 2024a; Luo et al., 2025). This results in an iterative scheme, where the updated student can be reused to guide further teacher-generated samples (Figure 1). Prior work has considered this paradigm by prioritizing incorrect student answers (Lee et al., 2024) and using `LLM-as-a-judge` scoring (Jiang et al., 2023b), but they do not draw upon the vast active learning and data selection literature. Instead, this paper advocates for the generation of data that are conditioned on samples that have been prioritized by an active learning algorithm. The resulting dataset enables more effective and data efficient finetuning of the SLM student model (see Section 5.4 for evidence supporting this claim).

Our work makes the following contributions:

- We provide a benchmark study for iterative synthetic data generation rooted in prior work on active learning and data selection. We carefully compare to static dataset generation—identical to uniform random sampling of prompts—to show improvements in data efficiency.

- We compare a range of methods for selecting samples for synthetic data generation, including those that favour uncertainty, diversity, or difficult/easy samples. We conclude that simple methods rooted in active learning, such as using the loss of the student's own prediction (Settles & Craven, 2008) are the most data efficient. In contrast, expensive and popular methods based on using an LLM to judge the difficulty and quality of data, i.e. `LLM-as-a-judge` (Zheng et al., 2023; Jiang et al., 2023b), underperform simpler and more general active learning alternatives.

- We show that synthetic data generation is to a certain extent "steerable"; the properties of the teacher generated synthetic data resemble those of the data selected by the student. For example, if the student selects "difficult" data—with a high loss—then the teacher also generates "difficult" data in aggregate.

## 2 PRELIMINARIES

**Notation.** We use $i$ to index a datapoint in a dataset and $j$ to index a token's position in the sequence. In our framework, learning happens iteratively, where synthetic samples are acquired from the teacher, the student trains on the new, larger dataset, and the process repeats. We use $t$ to index the iteration of iterative synthetic data generation. We denote question and answer pairs $z = (x, y)$, from a dataset of size $n$ drawn from a ground truth distribution $P$: $D_0 = \{z\}_{i=1}^n \sim P$. We use the terms "question" and "instruction" interchangeably for $x$, and "answer" and "response" interchangeably for $y$. The rationales or chain-of-thought steps (Wei et al., 2022) are incorporated into the answers $y$, however some datasets are comprised of answers without any chain-of-thought steps. A model $f_{\boldsymbol{\theta}}(\cdot)$ with parameters $\boldsymbol{\theta}$ generates an answer $\hat{y}$ given a question $x$: $\hat{y} = f_{\boldsymbol{\theta}}(x)$. Synthetic questions and answers are denoted $\hat{z} = (\hat{x}, \hat{y})$. Text is encoded into tokens, we denote $V$ as the vocabulary and each token is an indicator vector $\{0, 1\}^{|V|}$. SFT involves minimizing the next token prediction loss, the cross-entropy, over answer tokens given a question: $\mathcal{L}(z, \boldsymbol{\theta}) = -1/|y| \sum_{j=1}^{|y|} y_j \log f_{\boldsymbol{\theta}}(x, y_{<j})$. The model $f_{\boldsymbol{\theta}}(\cdot)$ autoregressively generates the next token $\hat{y}_j = f_{\boldsymbol{\theta}}(x, \hat{y}_{<j})$ in the sequence.

**Data Efficiency.** For a fixed number of samples, if better generalization performance can be achieved by training on one subset of a larger dataset than on another, the former can be considered more data efficient. Formally, let $P$ be the true data distribution over our data $z = (x, y)$. For a selec-

tion algorithm $\phi$ that produces a dataset $S_n^\phi = \{z_i\}_{i=1}^n \overset{\phi}{\sim} P$, model parameters $\boldsymbol{\theta}_n^\phi$ result from minimizing the loss over $S_n^\phi$. We define the performance, accuracy for example, on a single sample as

$$\text{perf}_\phi(z, \boldsymbol{\theta}_n^\phi) = \mathbf{1}\left\{y = f_{\boldsymbol{\theta}_n^\phi}(x)\right\}, \tag{1}$$

and the expected performance as

$$\text{perf}_\phi(n) = \mathbb{E}_{z \sim P} \mathbb{E}_{S_n^\phi \sim P}\left[\text{perf}_\phi(z, \boldsymbol{\theta}_n(S_n^\phi))\right]. \tag{2}$$

Assuming a monotonic increase in performance with $n$, for some target performance $\tau$, the sample complexity can be defined as

$$N_\phi(\tau) = \inf\left\{n : \text{perf}_\phi(n) \geq \tau\right\}, \tag{3}$$

which measures the smallest $n$ such that $\text{perf}_\phi(n) \geq \tau$. For a fixed architecture $f(\cdot)$, algorithm $\alpha$ is more data-efficient than algorithm $\beta$ at level $\tau$ only if $N_\alpha(\tau) < N_\beta(\tau)$ or if, for a fixed $n$, $\text{perf}_\alpha(n) > \text{perf}_\beta(n)$.

## 3 RELATED WORK

**Distillation.** Fitting models on synthetic datasets composed of pairs $z = (x, \hat{y})$ of sequences where $\hat{y}$ is produced by a teacher model conditioned on separately available prompts $x$—often referred to as distillation (Hinton, 2015)—has been shown to be extremely effective in improving capabilities of SLM student models (Taori et al., 2023; Peng et al., 2023; Team et al., 2024).

**Synthetic question and answer generation.** Going one step further, we can generate *both* questions *and* answers: $\hat{z} = (\hat{x}, \hat{y})$. SFT on synthetic question-answer pairs results in improved capabilities without being restricted by potentially small seed dataset sizes (Mitra et al., 2024). Much like in the distillation setting, generating a question-answer pair only requires prompting the teacher model with a seed data point (Liu et al., 2024a; Luo et al., 2025; Zeng et al., 2024).

**Selective question and answer generation.** Synthetic datasets are known to be compressible—synthetic samples filtered to have high `LLM-as-a-judge` (Chen et al., 2023) values or low student loss (Li et al., 2024), for example, can obtain the same performance as finetuning on the entire unpruned corpus. To remedy this inefficiency, rather than generating a large static synthetic dataset and then filtering, we can instead carefully select the seed data used to generate the synthetic samples, and hopefully produce fewer semantically similar sequences. This has been shown by prioritizing incorrect samples when querying the teacher, which is more data efficient than finetuning on the original corpus $D_0$ (Lee et al., 2024). `LLM-as-a-judge` selection is also known to be more data efficient than directly finetuning on a public benchmark synthetic datasets (Jiang et al., 2023b). This study includes `LLM-as-a-judge` scoring due to its widespread use.

### 3.1 ASSIGNING A VALUE TO DATA

**Active learning.** Our work makes use of ideas from active learning, which seeks to maximise data efficiency by iteratively identifying and prioritising informative samples for labelling (Settles, 2009; Settles & Craven, 2008). Classic strategies for active learning include model prediction disagreement (Freund et al., 1997; Houlsby et al., 2011), uncertainty (MacKay, 1992; Gal et al., 2017; Kirsch et al., 2019), and dataset summarization (Sener & Savarese, 2018; Mirzasoleiman et al., 2020; Coleman et al., 2019). Effective, contemporary methods trade-off between predictive uncertainty and sample diversity in a fashion that is commensurate with large neural networks (Ash et al., 2021; Saran et al., 2023). We consider language model-aligned variations of two popular methods for active learning: uncertainty sampling (Settles & Craven, 2008), a classic approach that favours predictive uncertainty, and BADGE, a more modern algorithm (Ash et al., 2019).

**Data selection.** Related methods aim to estimate the value of data to guide selection, typically using labelled dataset $(x, y)$. Data can be valued using Shapley values (Ghorbani & Zou, 2019), influence functions Koh & Liang (2017) or by matching training data to desirable evaluation datasets Just et al. (2023); Kessler et al. (2025); these methods have shown limited effectiveness for language

---

**Algorithm 1** Iterative synthetic data generation algorithm for question and answer datasets.

---

**Input:** Seed dataset $D_0$, test set $D_{\text{test}}$, train set $\hat{D}_{-1} = \{\ \}$, student $f_{\boldsymbol{\theta}}(\cdot)$, selection algorithm $\phi$.

1: **for** $t = 0, \ldots, T$ **do**
2:     Generate SLM predictions on $D_t$: $\{z_i = (x_i, \hat{y}_i)\}_{i=1}^n$ where $x_i \in D_0$ and $\hat{y} = f_{\boldsymbol{\theta}}(x)$.
3:     Select data subset: $\bar{D}_t = \phi(D_t)$.                  ▷ See Section 4.1 for details.
4:     Generate synthetic dataset: $\hat{D}_t = \text{Generate}(\bar{D}_t)$.      ▷ See Section 4.2 for details.
5:     SFT on $f_{\boldsymbol{\theta}}(\cdot)$ using $\hat{D}_t := \hat{D}_t \cup \hat{D}_{t-1}$ and evaluation on $D_{\text{test}}$.
6: **end for**

---

modelling. LLMs have been used to score data points (Zheng et al., 2023) and for selecting question-answer samples for SFT (Liu et al., 2024b; Jiang et al., 2023b; Chen et al., 2023). Still, it has been shown that LLMs scores exhibit biases that hinder their effectiveness in this setting (Xiong et al., 2024; Dorner et al., 2025; Panickssery et al., 2024). Alternative approaches use training loss or gradient norms with respect to student parameters as an estimate of learning progress (Loshchilov & Hutter, 2015; Katharopoulos & Fleuret, 2018; Jiang et al., 2019; Li et al., 2024; Mindermann et al., 2022; Evans et al., 2024; Dai et al., 2025). However, this has shown limited data efficiency for language models (Kaddour et al., 2023). These ideas have also been applied to efficiently distill knowledge from large teacher language models into smaller student models, to reduce the number of teacher evaluations by leveraging active learning uncertainty estimates (Zhang et al., 2024) and by simply identifying incorrect student responses (Liu et al., 2024a; Du et al., 2025). Reward models are commonly used to score and identify data points for SFT (Cao et al., 2023; Dubey et al., 2024). This work focuses on reward selection because of its popularity and prioritizing incorrectly answered student responses due to its simplicity.

## 4 ITERATIVE SYNTHETIC DATA GENERATION

The general iterative synthetic data generation process studied in this paper is shown in Algorithm 1 (Jiang et al., 2023b; Lee et al., 2024). We expand upon the algorithm's design choices in the next sections. Most of these methods can be thought of as explicitly scoring each sample with a value $\{s_i\}_{i=1}^n$ where $n = |D_0|$ and $D_0$ is the initial question-answer seed dataset. In these cases, we can select $m = |\bar{D}_t|$ points with the highest scores equivalent to selecting the "hardest" points, with the highest uncertainty for instance (described in the next section), which is sometimes called "argmax" selection $\bar{D}_t = \text{argmax}_m \{s_i\}_{i=1}^n$. For completeness, we ablate these decisions, for example instead selecting the "easiest" points with lowest uncertainty, and sampling proportionally to scores instead of using argmax selection (Appendix C.6). Concrete instantiations of selection strategies $\phi$ are outlined below.

### 4.1 SELECTION ALGORITHMS

**Uncertainty sampling.** A common method in the active learning literature is uncertainty sampling, which, for non-sequential classification models, prioritizes data for which the amount of probability mass on the most likely class predicted by the model is smallest (MacKay, 1992). In the sequential, Transformer-based setting, we can score a data point with the loss of the response tokens under the student $f_{\boldsymbol{\theta}}(\cdot)$ with parameters $\boldsymbol{\theta}$ as $\mathcal{L}(z_i, \boldsymbol{\theta})$ (Settles & Craven, 2008). When the targets used to produce a loss are the model's own generations, this score reflects an uncertainty in the produced sequence. Note that our setting gives us access to the ground-truth label associated with $x$ as well, and thus allows us to compute a true loss in a fashion commensurate with conventional model training (Loshchilov & Hutter, 2015). Interestingly, we find this to be less effective empirically than using the former, uncertainty-based approach.

**Reward scores.** Using the student's own generated sequence $\hat{y}$, a common method for scoring data is to obtain a prediction from a separate reward model $r(x, \hat{y})$. Resulting scores can be interpreted as the quality of the student's response, and indicative of its competence on questions of this sort in general. We are not limited to using the student's predictions, and can instead obtain a reward for the ground truth answer $y$ (Dubey et al., 2024). In this manner, rewards capture the difficulty of the data,

but this score has no dependence on the student model—we find that using $r(x, y)$ underperforms using $r(x, \hat{y})$ empirically for this reason.

**`LLM-as-a-judge` scores.** We can also leverage the reasoning ability of an LLM teacher model to score an SLM's predictions. This strategy asks the LLM teacher to score the detail, quality and correctness of the student's answer and reasoning with a value between $[1, 10]$. In particular, we use pairwise `LLM-as-a-judge` scoring which has been shown to be most effective (Zheng et al., 2023). Two separate answers are given for the teacher to decide which it prefers by providing scores for both: $s_i^t, s_i = \text{LLM}(\hat{y}_i^t, \hat{y}_i, x_i)$ where $\hat{y}_i^t = \text{LLM}(x_i)$ is teacher's answer, $s_i^t$ is the score for the teachers answer and $\hat{y}_i$ the student answer. This is an expensive scoring method since it requires the teacher to produce an answer in addition to scoring.

**BADGE.** Batch Active learning by Diverse Gradient Embeddings (BADGE) is a two-stage active learning algorithm. It first represents all candidate data using the last-layer gradient of the loss induced by treating the generated sequence as ground truth, $\nabla_{\theta_o} \mathcal{L}(\hat{y} = f_\theta(x))$, where $\theta_o$ are output-head parameters. BADGE then approximately samples from a $k$-DPP to identify gradients that are both high-magnitude and diverse (note that high-magnitude gradients are high-loss generations, suggesting high predictive uncertainty) (Ash et al., 2019). Like in uncertainty sampling, our setting allows us to use ground-truth target sequences, which would make these gradient representations of the sort used for optimization, but we found that using generated sequences resulted in better performance. Because the un-embedding layer of a Transformer is typically extremely large, we use a sparse random projection to efficiently reduce dimensionality while preserving geometric relationships (Johnson et al., 1984).

### 4.2 PROMPT-BASED SYNTHETIC DATA GENERATION

Selected data points $\bar{x}_i \in \bar{D}_t$ are added to a synthetic data generation prompt for the LLM teacher model to generate a synthetic question $\hat{x}_i$ (Xu et al., 2024; Mitra et al., 2024; Jiang et al., 2023b; Lee et al., 2024). The teacher is then prompted to produce chain-of-thought reasoning and a final answer for $\hat{y}_i$. We generate a synthetic data point $\hat{z}_i = (\hat{x}_i, \hat{y}_i)$ using $\hat{x} = \text{LLM}(\bar{x}_i)$ and $\hat{y}_i = \text{LLM}(\hat{x}_i)$. So $\hat{D}_t = \text{Generate}(\bar{D}_t) = \{\hat{x}_i = \text{LLM}(\bar{x}_i), \hat{y}_i = \text{LLM}(\hat{x}_i)\}_{i=1}^m$, where $\bar{x}_i \sim \bar{D}_t$. For the further details on the prompts used for each dataset see Appendix D.2.

## 5 EXPERIMENTS

This section empirically probes the data efficiency of iterative synthetic data generation against static data generation, and provides recommendations regarding which scoring and selection design choices improve efficiency. **We find that prioritizing difficult data, measured as the student's loss on its own generation, to be most data efficient.** We further provide probing experiments that give an exploration for this improved data efficiency. **On average, we show that synthetic data inherits some properties from the samples used to generate them.** If we choose difficult samples for the student—measured by a high loss or a low reward for example—the synthetic data is difficult as well, resulting in lower student accuracies on these generated samples than random selection. Both of these observations explain why selecting data prior to synthetic data generation results in synthetic data that has similar properties to our selected data.

At each iteration $t$ we use a given acquisition algorithm to select 1k samples $\bar{D}_t$ from $D_t$, before sending each to the teacher model to generate corresponding synthetic data $\hat{D}_t$. These data are appended to synthetic data from all previous iterations before reinitializing the student model and refitting its parameters.

### 5.1 DATASETS

This section presents results on four distinct reasoning datasets in conjunction with four different models. `GSM8k` is a popular mathematics dataset comprised of school level maths problems (Cobbe et al., 2021), which we use with a `Mistral-7B-Instruct-v0.3` student (Jiang et al., 2023a). Similarly, we include the more challenging `Math1-3` dataset (Hendrycks et al., 2021), which is comprised of 5 distinct levels of question difficulty—we use the easiest levels, 1 to 3, to finetune a

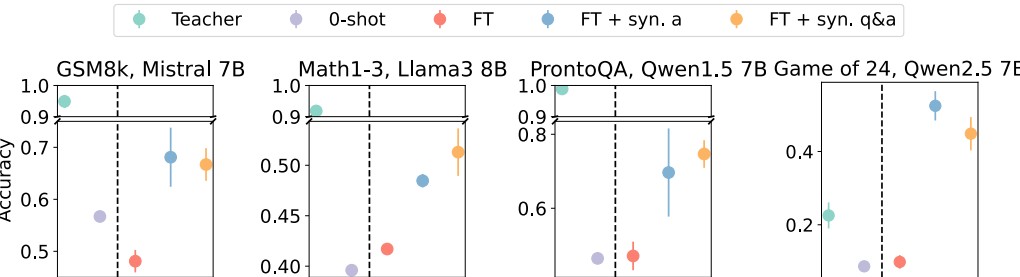

Figure 2: **SFT performance on** $1$**k data points for various datasets and SLMs.** We compare the effect of synthetic answer generation and synthetic question and answer generation to using the seed dataset, $D_0$ for SFT. 0-shot SLM and teacher performances are included for reference. All datasets use a `GPT-4o` teacher, for `Game of 24` we use a `GPT-o3-mini` teacher.

`Llama-3-8B-Instruct` student (Dubey et al., 2024). We further experiment with the logical reasoning dataset `ProntoQA` (Saparov & He, 2023), composed of synthetically generated chain-of-thought style reasoning questions, with a `Qwen1.5-7B-Chat` student. Finally we consider the `Game of 24` dataset, where a model is required to find the arithmetic operations to obtain $24$ given $4$ separate numbers. Here we use a `Qwen2.5-7B-Instruct` student (Qwen et al., 2025). More dataset details are provided in Appendix D.1.

For all datasets except for `Game of 24` we use prompt-based synthetic data generation with a `GPT-4o` teacher; see Appendix D.2 for our prompts. For `Game of 24` we use backward reasoning: if the answer is `13*8-10*8=24`, for example, we can construct a new question by setting two integers to variables `a*b-10*8=24` and solving to generate new questions (Jiang et al., 2024). We use a `GPT-o3-mini` teacher for backward reasoning, as it seems to produce better question-response pairs than `GPT-4o` (Appendix D.2.4).

## 5.2 Finetuning Setup

To enable new instruction-following capabilities we finetune our student on synthetic data $\hat{D}_t$, which are appended to synthetic data from all previous iterations $\hat{D}_{<t}$. For efficient training we adapt LoRA layers (Hu et al., 2022) after each iteration of acquiring data and fitting the model. We avoid warm starting SFT parameters from their pre-trained values and instead use a fresh, random reinitialization (Ash & Adams, 2020; Springer et al., 2025). We set the LoRA rank and alpha parameters to the same value (see Appendix B.1) and adapt all linear layers. For optimization we use Adam (Kingma & Ba, 2014), clamp the gradient norm to a maximum of $2.0$, and use a batch size of $24$ with $2$ gradient accumulation steps. The learning rate decays linearly with a warm up period of $15\%$ of the total number of epochs. For `Game of 24` we use a cosine decay learning rate schedule down to a minimum of 1e-9 (Ni et al., 2025). During optimization we perform checkpointing and load the checkpoint with the best performance on a held-out validation set after SFT. We search over learning rates, LoRA ranks and the number of training epochs on this held-out validation set (Appendix B). We use a single 80Gb A100 or H100 GPU for all experiments.

## 5.3 Algorithms

We consider a variety of selection algorithms to understand which are most sample efficient. Prior work has shown that prioritizing "hard" samples accelerates learning (Section 3.1), which we also find to be the case for iterative synthetic data generation (Appendix C.6). This approach prioritizes high-uncertainty data, measured as the model's loss on greedily decoded student generations, which we denote as "loss (high)" throughout this section. We also consider a low-reward selection algorithm ("rwd (low)"), also using the student's own predictions, which scores generations using an external model. We use a `Skywork-Reward-Llama-3.1-8B-v0.2` reward model which is the highest scoring 8b model on RewardBench (Lambert et al., 2025) at the time of writing.

We use Lion (Jiang et al., 2023b) as a baseline, which compares the student and teacher answer `LLM-as-a-judge` scores to categorize a data point as either hard or easy. All seed data are assigned into either an easy or a hard set before sampling equally from both. For completeness, we also consider

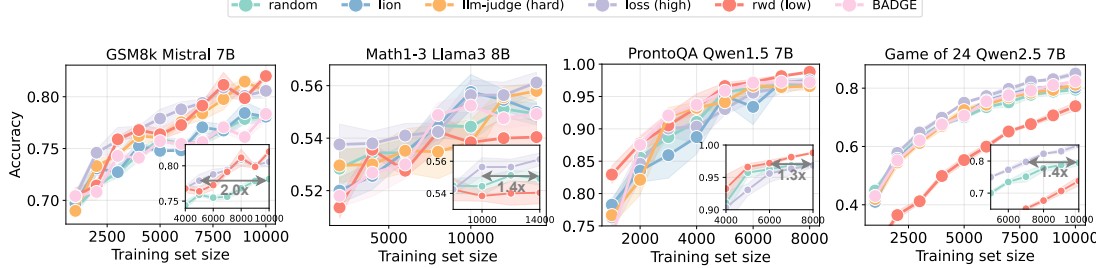

Figure 3: **Student performance over successive synthetic data iterations with growing training sets (Algorithm 1)**. Each inset plot shows the proportion of data random sampling requires for the same performance as the best active scorers for synthetic data generation.

a baseline that only samples from the hard set, denoted as `LLM-as-a-judge` (hard) (Jazbec et al., 2024). We use the same prompts for `LLM-as-a-judge` scoring as Jiang et al. (2023b).

We further consider prioritizing data with incorrect student answers, $s_i = \mathbf{1}\{\hat{y}_i \neq y\}$ as a proxy for prioritizing hard samples (Lee et al., 2024). In a similar spirit to Lion, we can instead sample evenly from correct and incorrect pools to maintain diversity in the synthetic data used for generation (Liu et al., 2024a). Since correct and incorrect selection requires a verifier and ground-truth answers we do not compare to other scoring methods which do not use label information, we place these supplementary results in Appendix C.1.

## 5.4 RESULTS

**Using synthetic data yields significant gains in student capabilities** compared to a fixed seed dataset of the same size (Section 5.4.1), additionally **iterative synthetic data generation is more data efficient than static generation** (Section 5.4.2). Note that static generation is equivalent to random sampling of prompts for synthetic data generation in our setting, since it is not conditioned on the current state of the student. Finally, we analyse our synthetically generated data to show that **it retains important properties of the original seed data,** (Section 5.4.3). Unless stated otherwise, results are a mean and standard deviation over 3 independent runs.

### 5.4.1 TRAINING ON SYNTHETIC DATA IMPROVES PERFORMANCE

**SFT on synthetic data results in significantly improved capabilities when compared to using the original seed dataset.** Figure 2 compares SFT performance on the seed data to synthetic data of equal size, showing a dramatic increase in performance across all datasets when doing SFT on synthetic question-answers pairs. In the same figure, we see large increases in performance when using synthetic answers $z_i = (x_i, \hat{y}_i)$ instead of seed answers $y$, likely due to better formatting and high quality chain-of-thought in synthetic answers. In `Game of 24` there is a small drop in performance when training on synthetic questions and answers versus only synthetic answers, showing that the generation of novel questions by the teacher yields some lower quality synthetic questions. Regardless, next we show how this enables us to scale dataset sizes efficiently.

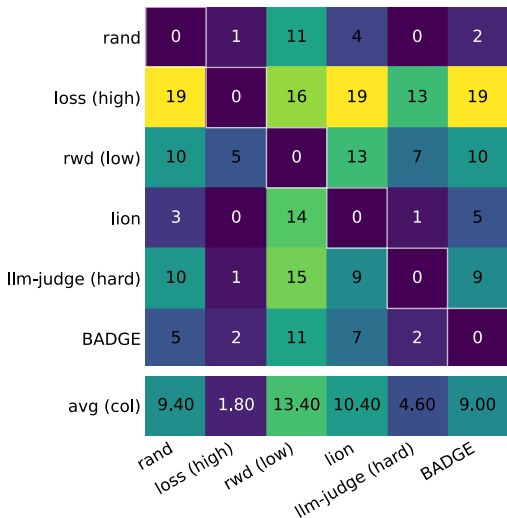

Figure 4: **Pairwise winrate over all datasets and methods.** $P_{ij}$ corresponds to the number of times algorithm $i$ outperforms $j$. Overall performance is shown in the last row (lower is better).

### 5.4.2 ITERATIVE GENERATION YIELDS MORE DATA-EFFICIENT RESULTS

**Active selection is more data efficient than random sampling for generating productive**

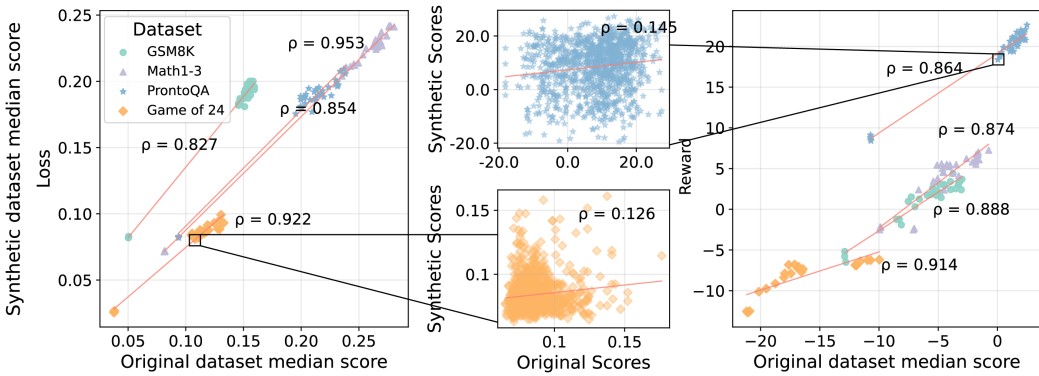

Figure 5: **The rank correlations between original and synthetic dataset scores from iterative synthetic data generation**. We plot student loss and reward scores and show Spearman's rank correlations ($\rho$) between dataset medians before and after synthetic data generation. We zoom in on relationships at an individual data-point level where there is low correlation between the original and synthetic data point scores (centre). The red line is the line of best fit to the data. All rank correlations are highly significant with a p-value of $p < 0.001$.

**synthetic data, resulting in better performance using fewer samples**. In Figure 3, we can see that random sampling underperforms when compared to certain active selection methods across all datasets. Among these, we find that simply prioritizing high-loss data consistently performs well.

To compare algorithms across all datasets we can aggregate results as a pairwise winrate matrix $\boldsymbol{P}$. We increment $\boldsymbol{P}_{ij}$ if $\mathbf{1}\{\hat{\mu}_i - \alpha \cdot \hat{\text{se}}_i > \hat{\mu}_j + \alpha \cdot \hat{\text{se}}_j\}$, where $\hat{\mu}_i$ is the sample mean and $\hat{\text{se}}_i$ is the standard error of the performance of algorithm $i$ for a dataset, for a particular dataset size, and $\alpha$ is the confidence level which we set to 1 (making it a 68% confidence interval). By summing the "wins" across the rows and normalizing we can understand how often algorithms are outperformed on average. Column-wise averages are shown in the last low, where lower is better, to understand which algorithm is more data efficient in total. We find that using random sampling is outperformed by various other methods that use the student model to guide synthetic data generation (Figure 4).

We can glean from Figure 4 that the best selection algorithm simply uses a high loss to select data that are difficult for the SLM. `LLM-as-a-judge` also tends to be effective, though by a smaller margin. Interestingly, BADGE and Lion, which both aim to select diverse data, do not perform much better than random sampling (Figure 4). It is worth mentioning that, because of the need to access a teacher model for scoring, `LLM-as-a-judge` is computationally demanding. Assuming that the cost of evaluating the teacher model dominates the cost of evaluating the student, a common assumption in the active learning literature and a reasonable assumption as the number of parameters of the teacher model can be 3 orders of magnitude larger than the student models we consider. Then, if we consider the total number of teacher input and output tokens as a budget instead of the number of generated samples, Lion and `LLM-as-a-judge` (hard) are far more expensive than other methods (Figure 6).. This additional compute is better allocated towards simply generating more synthetic data with a cheaper and more effective selection strategy.

Reward scoring also requires an external model, but because we can use a reward model that has the same number of parameters as our student, calls to the reward model are generally less expensive than to a teacher—we opt to not treat them in the same way and do not count the number of input tokens to the reward model in Figure 6. Overall random selection requires between 33% to 100% more SFT data to obtain the same performance as the best selection methods across all datasets (Figure 3). For 2/4

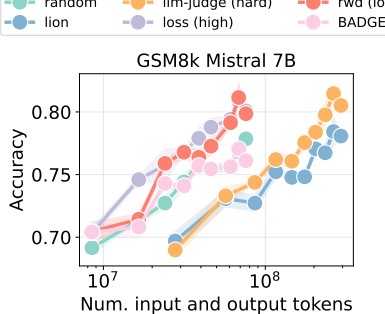

Figure 6: **Iterative synthetic data generation learning curves on `GSM8k`: student performance versus the number of teacher input and output tokens**. The total number of input and output tokens are a proxy for the amount of compute used by the teacher for various selection methods.

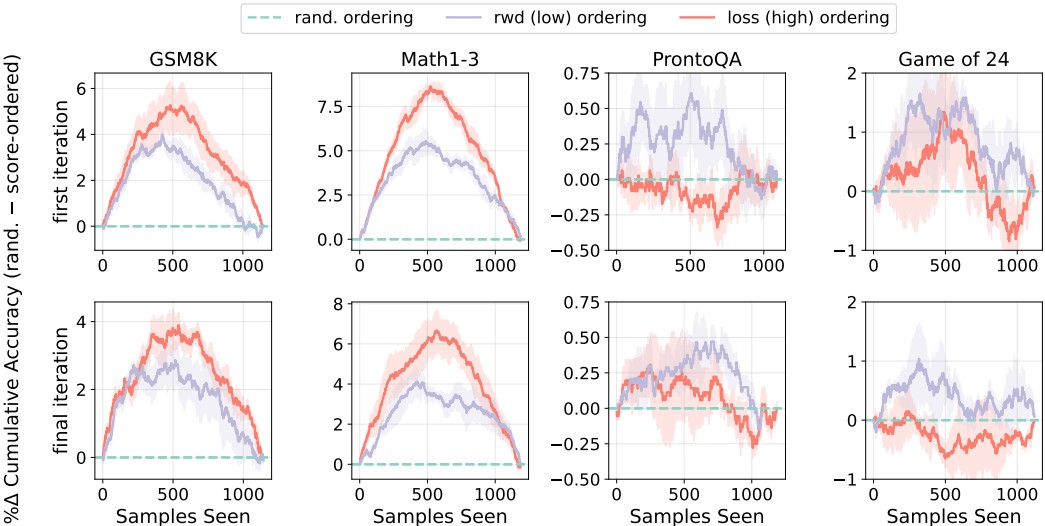

Figure 7: **The percentage difference in synthetic data cumulative accuracy when using random sampling minus score ordering: high to low loss and low to high reward**. For each original data point we score it using the student model from the first and final iteration of iterative synthetic data generation (rows). See Figure 11 for the cumulative accuracies for individual replicates.

of these datasets iterative synthetic data generation using the loss on the student's own predictions leads to more data efficient results compared to prior works that perform SFT using similarly sized datasets on the same SLM. See Appendix C.2 for comparisons.

### 5.4.3 ON THE FIDELITY OF SYNTHETIC DATA TO ITS ORIGINAL DATA

**Original dataset scores are retained by synthetic data.** When we measure the median score over the entire selected dataset and the corresponding median over the synthetic dataset we find a very high rank correlation over the course of iterative synthetic data generation. This shows that at a dataset level the order of scores—uncertainty (measured by the loss) or data quality (measured by the reward)—from the iterative synthetic dataset curriculum are preserved (Figure 5). In spite of this, synthetic data generation is an uncontrollable and noisy process which perturbs data by rephrasing, complicating or simplifying and adding chain-of-thought rationales. This is manifested when we plot the scores of the individual data points before and after synthetic data generation (Figure 5, centre). We notice that the individual data point rank correlations are low but significantly greater than zero across all datasets (Appendix C.3).

**Prioritizing "difficult" data results in more difficult synthetic data than random sampling.** We compare the cumulative accuracy of synthetic data ordered randomly minus the cumulative accuracy of synthetic data ordered with a high to low original loss and low to high reward in Figure 7. For GSM8k and Math1-3 the difference in the cumulative accuracies is significantly greater than 0. So prioritizing data according to these scores results synthetic data with lower student accuracies than random sampling. The synthetic data is "harder" using these active learning approaches and these "hardness" qualities are integrated into the synthetic data the teacher generates. This is also seen for the first iteration for the Game of 24 dataset. In contrast, at the final iteration of the Game of 24 dataset and for the ProntoQA dataset the student obtains high accuracies on the synthetic data so we cannot see any difference in cumulative accuracies. Using low reward selection produces more difficult samples compared to random for the Game of 24 dataset (Figure 7), and results in poorer SFT performance than random however (Figure 3). This is because we observe that the reward scorer introduces other biases; it is biased to long student answers and so biased toward a narrow stylistic set of student responses, as observed in other work (Shen et al., 2023; Bu et al., 2025). Moreover each selection algorithm has its own selection biases which results in different synthetic dataset token distributions, see Appendix C.5. See Figure 11 for the original cumulative accuracies used for calculating the percentage differences in Figure 7.

## 6 LIMITATIONS

**Iterative synthetic data generation for finetuning.**   We only consider SFT, we do not consider efficient synthetic data generation to accelerate training for RLHF, continual pre-training (Yang et al., 2025) or pre-training (Maini et al., 2025), for instance. These are promising directions of future work.

**The limits of the teacher.**   We assume that the teacher is able to generate high quality questions and answers. For `GSM8k`, `Math1-3` and `ProntoQA` the teacher performance is high and so we assume $\hat{z}_i$ is correct. For `Game of 24` we rely on backward reasoning (specific to arithmetic) and a verifier to assess the teacher's synthetic data. We have yet to test the limits of prompt-based synthetic data generation in settings where teacher capabilities fall short.

**Data generation is noisy.**   We can obtain improved student capabilities using iterative synthetic data generation. However, synthetic data generation is a noisy process where we show that properties of the selected datasets are preserved (Section 5.4.3). However, it is not clear how we can guarantee that synthetic data retains desirable properties from the seed dataset. For example, reward scoring performs poorly for the `Game of 24` since it is biased by long student answers despite also selecting low quality student responses for synthetic data generation as required. We have presented an initial study of the "steerability" of synthetic data generation. However the ability to add further desirable properties is left for future work.

## 7 CONCLUSION AND DISCUSSION

Synthetic data are extremely effective for finetuning SLMs, enabling substantial capability improvements. In this work, we focus on supervised finetuning with synthetic data. We demonstrate that iterative synthetic data generation is the most effective strategy for fine-tuning SLMs under a fixed training data budget. By adapting teacher generation to the evolving state of the student model, this approach creates a natural curriculum that consistently outperforms static synthetic datasets in both performance and data efficiency. Furthermore, in line with Occam's razor, we find that simple data selection methods, such as prioritizing hard samples with high loss, outperform complicated and expensive `LLM-as-a-judge` based methods. We also find that the student can steer the synthetic data generation since firstly dataset level statistics are preserved by synthetic data generation. Secondly, prioritizing hard samples in turn produces hard synthetic samples, that when trained on results in improved student performance. As a result, the synthetic data retains the characteristics of the student selected data, enabling additional student capabilities after finetuning.

## 8 REPRODUCIBILITY STATEMENT

Reproducibility goes to the heart of our study of different selection algorithms for data efficient synthetic data generation. In our study, all the results are stated as means and standard errors over 3 independent replicates. This has been done in an effort to encapsulate the variance arising from the datasets we use and our experimental setup, and ensures that the performance differences arise due to the choice of selection methods rather than random variation. As a result we use uncertainties to weight our claims resulting a more reproducible study.

We will release source code upon publication.

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

# Appendix

## Table of Contents

## A   THE USE OF LARGE LANGUAGE MODELS

We used a coding assistant to help implement and debug our experiments. Regarding paper writing we used LLMs for finding related work, to assist with grammar queries, and to assist in generating the figures in the paper.

## B   ADDITIONAL EXPERIMENTAL SETUP DETAILS

We introduce additional details of our experimental setup from Section 5.3. We outline the hyper-parameter grid search for SFT below.

### B.1   LoRA HYPERPARAMETER TUNING DETAILS

| Model | Dataset | LoRA Rank | Learning Rate | Epochs |
|---|---|---|---|---|
| Mistral-7B-Instruct-v0.3 | GSM8k seed | 32 | 1e-4 | 10 |
| Llama-3-8B-Instruct | Math1-3 seed | 32 | 1e-6 | 13 |
| Qwen1.5-7B-Chat | ProntoQA seed | 32 | 1e-5 | 13 |
| Qwen2.5-7B-Instruct | Game of 24 seed | 16 | 1e-5 | 13 |
| Mistral-7B-Instruct-v0.3 | GSM8k synthetic | 32 | 1e-4 | 10 |
| Llama-3-8B-Instruct | Math1-3 synthetic | 64 | 1e-4 | 13 |
| Qwen1.5-7B-Chat | ProntoQA synthetic | 32 | 1e-5 | 13 |
| Qwen2.5-7B-Instruct | Game of 24 synthetic | 16 | 5e-4 | 30 |

Table 1: Optimal hyper-parameters for LoRA fine-tuning for all seed and synthetic datasets.

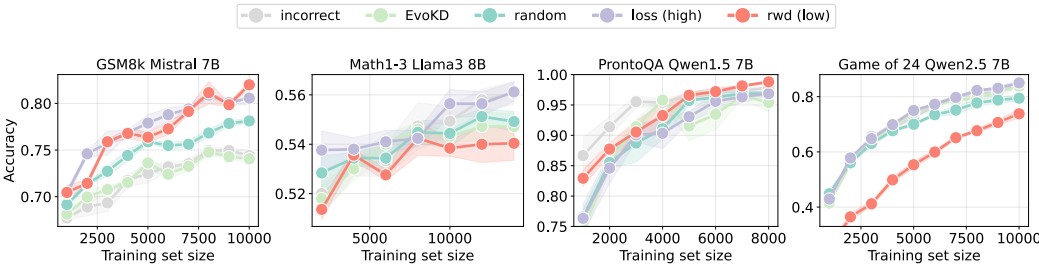

Figure 8: **Iterative synthetic data generation learning curves, showing student SFT performance after training on synthetic data of increasing size with incorrect and EvoKD data prioritization**. Each consecutive increase in dataset size corresponds to an iteration of iterative synthetic data generation (Algorithm 1). Learning curves are across various dataset student model pairs. Curves are an average and standard error over 3 replicates.

To obtain the best hyperparameters for our seed $D_0$ and synthetic datasets $\hat{D}_t$, we sweep through a grid of learning rates, number of training epochs and LoRA rank hyper-parameters using 1k question-answer pairs from the original seed dataset and 1k question-answer pairs synthetically generated by the teacher model. Refer to Table 1 for the optimal hyperparameters.

## C  ADDITIONAL RESULTS

We introduce additional results that support the claims in our main paper. In Appendix C.1, we introduce the results of prioritizing synthetic data generation using incorrect student predictions (Lee et al., 2024) and an even distribution of correct and incorrect student data (Liu et al., 2024a). We do not include these results in the main paper for comparison since they require the ground truth answer $y$ for scoring unlike the other active scoring methods considered (Section 5.4.2). In Appendix C.2, we compare iterative synthetic data generation with comparable SFT methods from the literature.

In Appendix C.3, we analyse the workings of synthetic data generation to show that despite introducing noise, the synthetic data retains the scores of the original selected seed data in aggregate. Furthermore, in Appendix C.4, we study how the synthetic datasets which are prioritized by low reward and high loss selection algorithms result in more difficult synthetic datasets since we observe lower student accuracies. In Appendix C.5, we show how the different scorers produce synthetic datasets with different token frequency distributions. These observations explain why selecting data prior to synthetic data generation results in data that has similar properties to our selected data and therefore enhanced student performance upon finetuning. Finally, in Appendix C.6, we compare various design choices for iterative synthetic data generation (Algorithm 1).

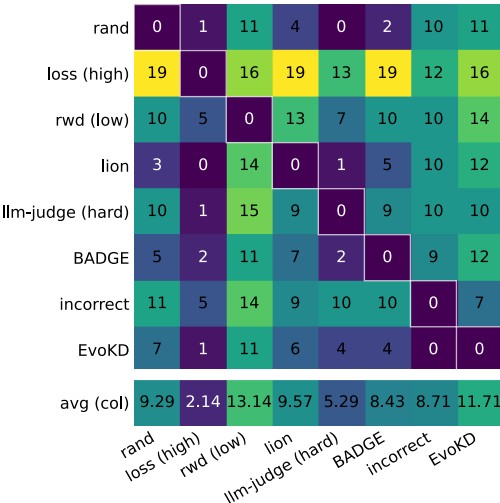

Figure 9: **The pairwise win rate matrix over all datasets and all methods including incorrect prioritization**. Element $P_{ij}$ corresponds roughly to the number of times algorithm $i$ outperforms algorithm $j$ including results of incorrect student answer prioritization (Lee et al., 2024) and even correct and incorrect answer prioritization (Liu et al., 2024a). Column-wise averages at the bottom display overall performance (lower is better).

### C.1  PRIORITIZING INCORRECT SAMPLES

**Prioritizing incorrect student predictions yields strong performance on all but one of the datasets we consider.** A simple data point scoring mechanism is to assign a $\{0, 1\}$ score for an incorrect or correct answer from the student model. This scoring mechanism requires a

| Dataset | Method | LLM | SFT Dataset Size | Performance |
|---------|--------|-----|------------------|-------------|
| GSM8k | Teacher | GPT-4o | n/a | $94.9 \pm 1.1$ |
|  | Orca-Math (Mitra et al., 2024) | Mistral-7B-Instruct-v0.3 | 10k | 70.2 |
|  | OpenMathInstruct (Toshniwal et al., 2024) | Mistral-7B-Instruct-v0.3 | 1.8M | 80.2 |
|  | Iterative Synthetic Data Generation (ours) | Mistral-7B-Instruct-v0.3 | 10k | $80.6 \pm 1.2$ |
| Math1-3 | Teacher | GPT-4o | n/a | $91.8 \pm 0.7$ |
|  | Iterative Synthetic Data Generation (ours) | Llama-3-8B-Instruct | 10k | $56.1 \pm 0.9$ |
| ProntoQA | Teacher | GPT-4o | n/a | $98.9 \pm 0.4$ |
|  | Iterative Synthetic Data Generation (ours) | Qwen1.5-7B-Chat | 8k | $96.9 \pm 0.8$ |
| Game of 24 | Teacher | GPT-o3-mini | n/a | $22.6 \pm 1.8$ |
|  | UFT (Ni et al., 2025) | Qwen2.5-7B-Instruct | 13.7k | $30.2 \pm 2.1$ |
|  | Iterative Synthetic Data Generation (ours) | Qwen2.5-7B-Instruct | 6k | $85.0 \pm 1.3$ |

Table 2: **Iterative synthetic data generation performs comparably to state-of-the-art SFT methods on certain datasets**. The results of iterative synthetic data generation using high loss selection, as this selection method performs the best overall. We compare only to methods that use the same LLM and omit work that relies on larger datasets to achieve higher performance, as we cannot determine whether such gains stem from better techniques or simply from increased data. All SFT methods report the amount of data used for SFT. We report a mean and standard error over multiple seeds for our work, however some baselines only report a single seed.

verifier or the ground truth answer $y$ and so is not directly comparable to the active scoring methods we consider that do not require the ground truth answer for scoring (Section 5.3). Regardless, we show the results of performing iterative synthetic data generation by prioritizing incorrect samples in Figure 8. For GSM8k this method severely underperforms other prioritization methods and random sampling. For Math1-3 and Game of 24 incorrect student answer prioritization is as data efficient as high loss scoring which is the most data efficient method identified in Section 5.4.2. For the ProntoQA dataset incorrect answer prioritization obtains results on par with the best scoring methods if not better results for certain dataset sizes $n$. Considering a pairwise win-rate (described in Section 5.4.2) we can see from the row for incorrect prioritization that it is more data efficient in many instances with a high number of "wins" versus other methods. However at the same time looking at the corresponding column it is outperformed by many of the other methods in particular high loss and low reward selection due to its poor performance on the GSM8k dataset so it results in a poor overall score in the final row (Figure 9). Overall it is a simple method and has the possibility of obtaining strong capabilities and being more data efficient than random sampling in certain domains.

**EvoKD underperforms random sampling and other active selection methods.** Similar to Lion, which samples evenly from easy and hard data pools as determined by LLM-as-a-judge scores, we can sample data evenly from correct and incorrect student predictions for synthetic data generation (Liu et al., 2024a). Evolving Knowledge Distillation (EvoKD), denoted as "EvoKD" in Figure 8, can be viewed as a diversity-based sampling approach for synthetic data generation. It achieves performance comparable to incorrect-data prioritization on GSM8k and Game of 24, but underperforms it on Math1-3 and ProntoQA. For GSM8k, EvoKD shares the same pathologies as promoting incorrect samples, they both underperform random sampling. EvoKD also underperforms methods that explicitly promote difficult samples (Figure 9), since it promotes hard samples through incorrect prioritization while simultaneously including easy samples to preserve the original data distribution. Overall, diversity-based criteria underperform approaches that emphasize difficult samples across the methods and datasets we study.

## C.2 COMPARING TO OTHER SFT METHODS

**Iterative synthetic data generation obtains comparable results to state-of-the-art SFT methods on certain datasets.** Table 2 compares the results of iterative synthetic data generation with high-loss selection to prior works in SFT which use the same LLM and similar dataset sizes. In our definition of data efficiency (Section 2), we can only properly compare against baselines that use the same model and that perform SFT on datasets of the same size, or if a baseline has a lower performance on a larger dataset size. Then we can conclude whether our method or the baseline is more data efficient, as defined in Section 2. If a baseline has better performance with a larger dataset size, then it is not possible to say whether the baseline we are comparing against or our method

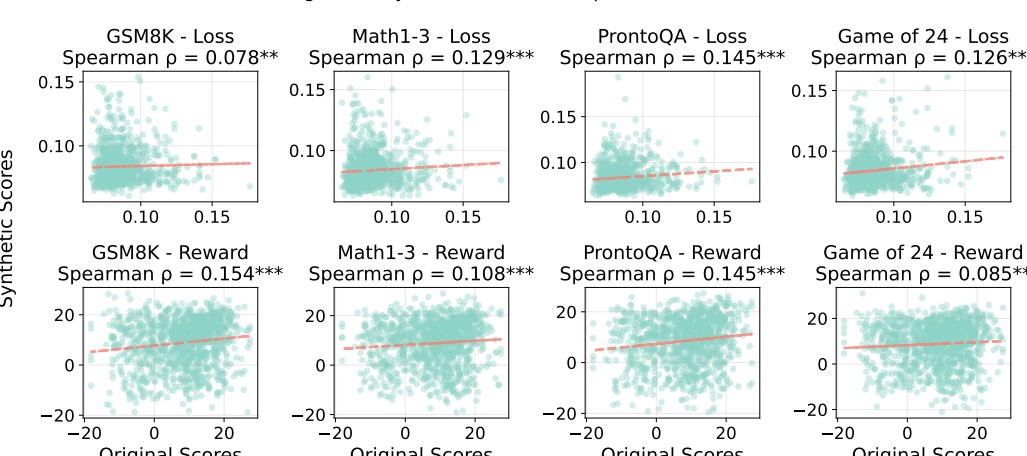

Figure 10: **The scores for individual datapoints before and after 1 step of synthetic data generation.** We consider the loss and reward of the student's predictions and look at the individual data points scores across all datasets. The Spearman correlation measures the rank correlation before and after synthetic data generation. The red line shows the line of best fit to the data. The number of asterisks denotes the rank correlation's p-value: *** indicates $p < 0.001$.

is more data efficient without scaling to the same dataset sizes. Since we cannot disentangle the performance improvements due to data quality or to increased dataset sizes. For `GSM8k` our work is more data efficient when compared to Orca-Math Mitra et al. (2024). Also for `Game of 24` our method outperforms state-of-the-art SFT baselines that use a `Qwen2.5-7B-Instruct` LLM Ni et al. (2025). For the `Math1-3` and `ProntoQA` datasets we did not find a comparable SFT methods to compare data efficiency with.

### C.3 SYNTHETIC DATA GENERATION PRESERVES PROPERTIES OF THE SELECTED DATA

**At the dataset level synthetic data generation preserves properties of the original seed dataset**. We score the selected seed dataset and take a median over scores and compare to the median score over the resulting synthetic data. If we do this for all iterations, we observe a very high rank correlation between median scores in Figure 5. **This indicates that the scores across the iterative synthetic data generation curriculum are similar before and after synthetic data generation**.

When we look at the scores over *individual* data points and consider the score of a selected data point and the corresponding score of the synthetically generated datapoint, then we find there is a low but significantly greater than 0 rank correlation between reward and loss scores for all datasets Figure 10.

These two observations are consistent: synthetic data generation is preserving distributional factors such as dataset uncertainty (as measured by the loss over student predictions) and dataset quality (as measured by the reward over student predictions). But the noise from prompt-based synthetic data generation means that there is a low but significant correlation between scores at an individual data point level.

### C.4 PRIORITIZING DIFFICULT DATA CREATES DIFFICULT SYNTHETIC DATA

**The teacher produces difficult synthetic data when hard samples are prioritized by the student.** We score seed data according to its loss or reward and then generate corresponding synthetic data. We obtain the cumulative accuracy of the synthetic data ordered by the original data scores. A random ordering corresponds to random sampling, while ordering the cumulative accuracy according to a high to low loss or low to high reward corresponds to prioritizing "difficult" data as we do in iterative synthetic data generation. For random sampling the cumulative accuracy versus the amount of data seen so far follows a diagonal line (Figure 11).

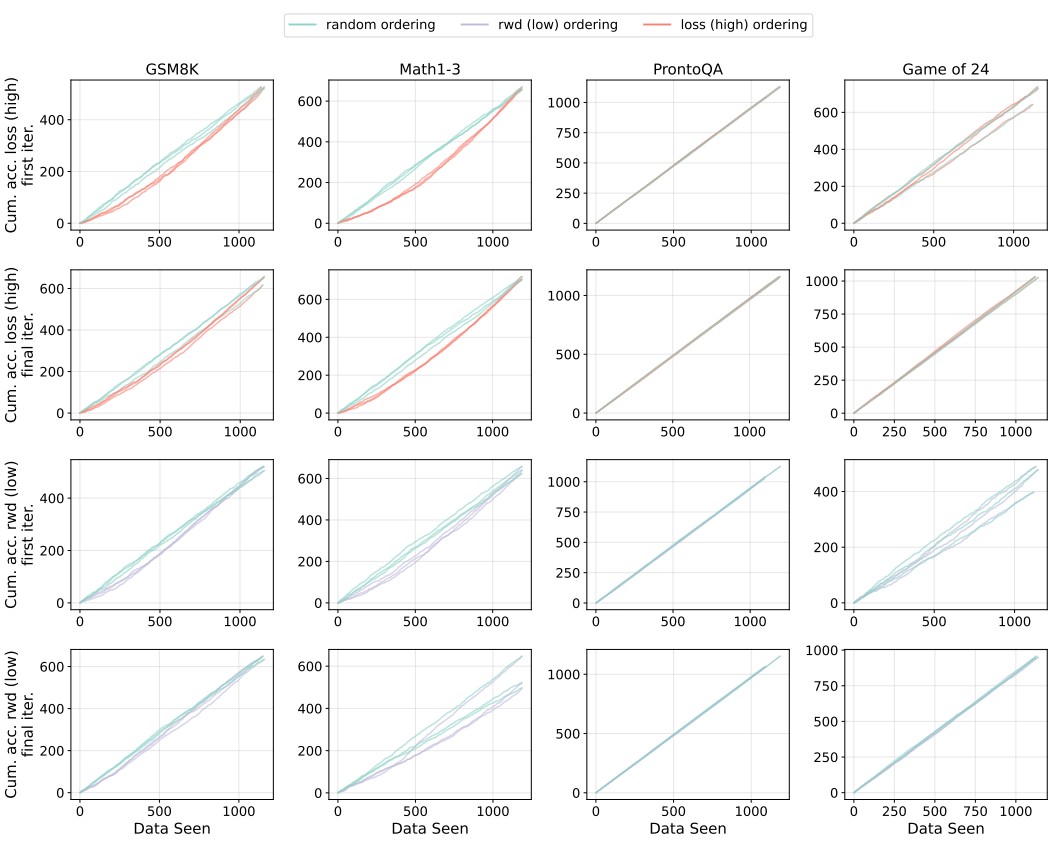

Figure 11: **The synthetic data cumulative accuracies when using random sampling and score ordering: high to low loss and low to high reward**. For each original data point we score it using the student model from the first and final iteration of iterative synthetic data generation (alternating rows). Then we generate a synthetic data point. We compare the cumulative accuracy over the synthetic data when ordering data randomly versus ordering according to the loss and reward scores. We plot individual replicates as individual lines.

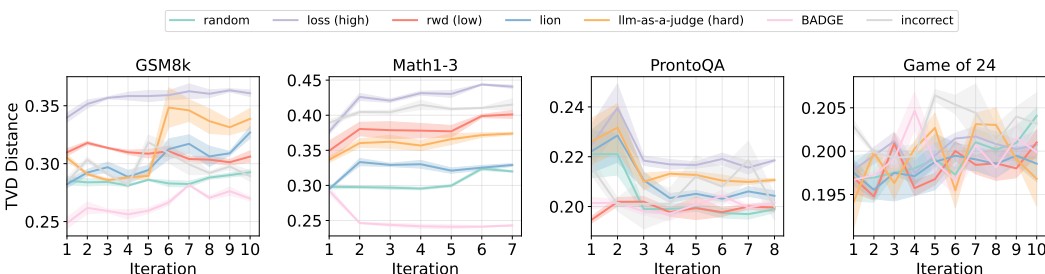

Figure 12: **The total variation distances between token distributions of our synthetic data and the original seed datasets.** We observe differences in the token distributions over the course of iterative synthetic data generation across for data selection algorithms, indicating differences in the synthetic datasets arise due to the different selection algorithms used.

We plot the cumulative accuracy curves for synthetic data ordered from high to low original data loss (loss (high) ordering) in the first two rows and by low to high original data reward (rwd (low) ordering) in the final two rows of Figure 11. For `GSM8k` and `Math1-3` the cumulative accuracy curves for synthetic data ordered using high to low original data loss and low to high reward are below random sampling so prioritizing data according to these scores results synthetic data that the student gets lower accuracies versus random sampling. The synthetic data is "harder" using these active learning approaches and these "hardness" qualities are integrated in the synthetic data the teacher generates. This is also seen for the first iteration for the `Game of 24` dataset for both scorers. In contrast, in the final iteration the student is able to get a high accuracy on the synthetic data and so it is difficult to see any difference between random ordering and prioritizing according to a high loss or low reward. This is also the case for the `ProntoQA` dataset, for the first iteration we see high student accuracies for the synthetic data making comparison versus random sampling difficult, despite the reward scorer obtaining better performance than random on the `ProntoQA` dataset (Figure 8).

To obtain the cumulative difference plots presented in the main body of this manuscript in (Figure 7), we simply take the vertical distances between corresponding random sampling cumulative accuracies and the scorer cumulative accuracies in Figure 11 and aggregate across all replicates to obtain means and standard errors.

## C.5 DIFFERENT SELECTION ALGORITHMS HAVE THEIR OWN SELECTION BIASES

**The different selection algorithms we consider manifest as differences in the synthetic dataset distributions.** When we compare the synthetic datasets to the original seed datasets over the course iterative synthetic data generation, then differences between selection algorithms are evident by looking at the token distributions in Figure 12. In particular, we measure the difference between two token distributions using the total variation distance (TVD): $\text{TVD}(P_{D_0}, P_{\hat{D}_t}) = \frac{1}{2}\sum_{x \in V}|P_{D_0}(x) - P_{\hat{D}_t}(x)|$ where $x$ is a token in the vocabulary $V$ and $P$ is the empirical token distribution. The token distribution $P$ can be thought of as a histogram where the bin size is the normalized frequency of the token in the dataset. This distance is essentially looking at the absolute differences in token counts between two datasets. When measuring the TVD between synthetic datasets and the original seed dataset prior to selection, $D_0$. We can see that the distance varies between different selection algorithms which shows that there are differences in the synthetic datasets at a token distribution level. The `Game of 24` dataset is the sole case where the selection algorithms yield almost indistinguishable TVDs, as its questions and answers draw from a highly restricted token range to compute 24 from four numbers using basic arithmetic operations. This points to there being distributional differences between synthetic datasets of different selection algorithms and thus shows that the selection algorithms manifest in different synthetic datasets with different properties over the course of iterative synthetic data generation. These distributional differences lead to performance differences between different selection algorithms which have been studied in the main results (Figure 4).

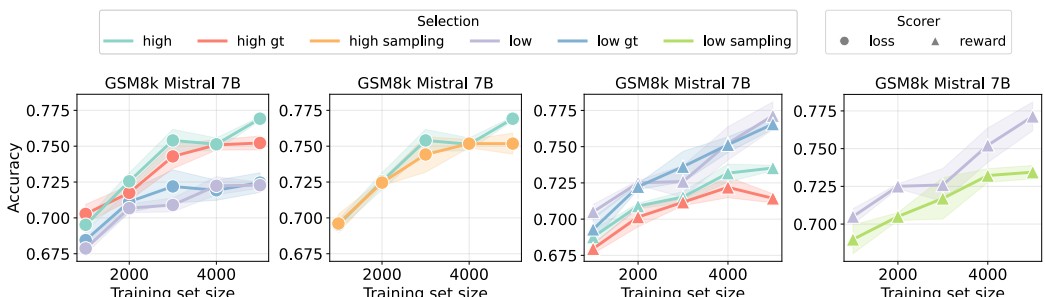

Figure 13: **Performance of iterative synthetic data generation on various data scoring and selection options**. We train on 1k data points at each iteration with a `Mistral-7B-Instruct-v0.3` student on `GSM8k`. We compare prioritizing "difficult" or "easy" data points with a high or low loss or reward. We compare using ground truth answers $y$ to the student's own predictions $\hat{y}$ and using argmax selection against sampling.

### C.6 ON THE DESIGN CHOICES FOR ITERATIVE SYNTHETIC DATA GENERATION

**Argmax selection, rather than sampling, results in the best SFT performance.** In Figure 13, we compare various data prioritization design choices. The performance for scorers that prioritize data where the student answer is the most uncertain (high loss) or worse quality (low reward) results in the best performance when compared to data for which the model is confident (low loss) or is of better quality (high reward). Furthermore, we compare whether using the ground truth answer $y$ (denoted "gt" in Figure 13) or the student's own prediction $\hat{y}$ is more data efficient. We can see worse performance when computing scores with the ground-truth answer for the loss scorer, while scoring with the reward model results in equal SFT performance. **There is no benefit to using the ground truth answers over the student's predictions for scoring data.**

Along the same lines we evaluated using ground-truth answers for the BADGE active learning algorithm (Ash et al., 2019) and found little benefit when compared to using student predictions (Figure 15). Moreover, using student predictions is consistent with the original intent of the BADGE algorithm. For these reasons, the results in the main paper make use of its implementation that relies on student predictions to obtain gradient embeddings.

Finally, we compare selection methods: argmax selection and sampling and can see lower SFT performance when using sampling (labelled with "sampling" in Figure 13). We sample $m$ points by sampling from a distribution proportional to these scores: $\bar{D}_t \overset{m}{\sim} \mathrm{softmax}(\{s_i\}_{i=1}^n)$. We found poor performance when sampling because sampling from the softmax distribution of loss or reward scores results in a similar distribution of scores for selected data as if we performed random sampling. Moreover, if we select $m = 1k$ data points from the `GSM8k` seed dataset and look at the distribution of loss scores via sampling for the highest and lowest 1k scoring data, then the distributions are indistinguishable to the naked eye. Argmax selection however produces distinct distributions (Figure 14).

## D DATASET FURTHER DETAILS

In this section we provide in depth details on the datasets used in our experiments together with the dataset sizes used throughout our empirical study of iterative synthetic data generation (Appendix D.1). Also we provide the teacher prompts used for synthetic data generation (Appendix D.2).

We introduce the seed question and answer datasets $D_0$. The validation and test sets are taken from the original seed datasets as opposed

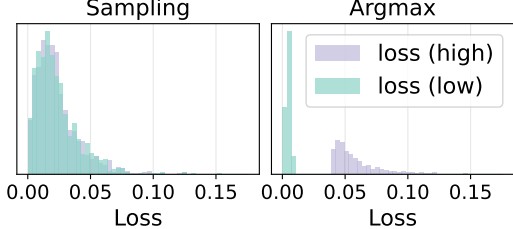

Figure 14: **Distribution of losses for different sampling methods**. We select 1k according to a high or low loss sampling (left) and argmax selection (right) for `GSM8k` and can see almost no difference when using sampling.

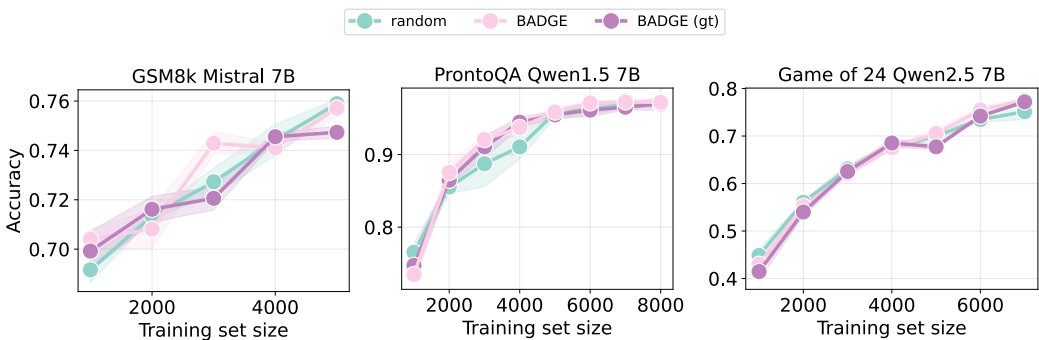

Figure 15: **Performance of design choices for BAGDE on various data sets**. We compare using the student's own predictions for producing gradient embeddings or using the ground truth answers (gt) for computing the gradient embeddings for BADGE, similar to gradient-based optimization.

to using synthetic data. The train sets $\hat{D}_t$ are synthetically generated. We summarize the datasets sizes in Appendix D.1. Unless otherwise stated we use a `GPT-4o` teacher. We prompt the teacher with few-shot examples from $D_0$ to generate a new synthetic questions (Liu et al., 2024a). For all datasets we throw away similar synthetic questions if the rouge-score (Lin, 2004) with respect to all previously generated questions is above $0.7$ (Jiang et al., 2023b).

**GSM8k.** We perform SFT on a `Mistral-7B-Instruct-v0.3` (Jiang et al., 2023a) student on school level mathematics questions (Cobbe et al., 2021). We use an external language model `gpt4o-mini` to assess whether the student's answer is equivalent to the ground truth answer, in a similar manner to Mitra et al. (2024), see Appendix D.3 for prompting details. We take $748$ question-answer pairs from the test set as a validation set and use $500$ question-answer pairs as a test set[*].

**Math1-3.** We finetune a `Llama-3-8B-Instruct` (Dubey et al., 2024) student on the competition math dataset (Hendrycks et al., 2021) which consists of more difficult math questions[†]. The dataset is classified into $5$ levels of question difficulty. We use the easiest levels $1$ to $3$ and pick $500$ question-answer pairs from the test set for validation. We assess the correctness of an answer by matching the solution to the regular expression `\boxed{(\d*)}`. The dataset is also categorized by the type of mathematics question: geometry, algebra etc. We use the category in our synthetic data generation prompt.

**ProntoQA.** The questions are synthetically generated logical chain-of-thought style reasoning questions with boolean answers (Saparov & He, 2023). We perform SFT with a `Qwen1.5-7B-Chat` student model. We use an external language model `gpt4o-mini` to assess whether the student's reasoning steps are correct and whether the student answer is equivalent to the ground truth answer like for `GSM8k`, see Appendix D.3 for details. We use $300$ question-answer pairs as a validation set and the remaining $200$ as a test set[‡].

**Game of 24.** We use a `Qwen2.5-7B-Instruct` (Qwen et al., 2025) student for SFT on the task of using $4$ numbers to obtain the number $24$ by finding which basic arithmetic operations are needed[§]. Each question can have multiple solutions, we treat each solution as a separate data point. We use backward reasoning to synthetically generate new questions (Jiang et al., 2024) and use `GPT-o3-mini` as a teacher model (qualitatively this produces better questions than `GPT-4o`). In backward reasoning if the answer is `13*8-10*8=24`, for example, we can construct a new

---

[*]`https://huggingface.co/datasets/openai/gsm8k`

[†]`https://huggingface.co/datasets/hendrycks/competition_math`

[‡]We use`https://huggingface.co/datasets/renma/ProntoQA` for validation and testing, as a train set we use `https://huggingface.co/datasets/longface/prontoqa-train` like in (Huang et al., 2024), questions are distinct between these two ProntoQA datasets.

[§]`https://huggingface.co/datasets/nlile/24-game`

question by setting two integers to variables `a*b-10*8=24` and solving to generate new questions and answers (Jiang et al., 2024). We verify that the backward reasoned final answer evaluates to $24$ and that it uses the $4$ numbers in the question. We use `GPT-4o` to then generate reasoning steps to obtain the final backward-reasoned answer. We assess the correctness of the student's final answer by matching the regular expression in `\boxed{}` and that the extracted answer evaluates to $24$ and checking that all numbers in the question are used once. Synthetic questions are not checked for rouge-score overlap since the set of tokens required to make questions and answers is a small subset of the vocabulary.

| Dataset | Seed Size | Validation Size | Test size |
|---|---|---|---|
| GSM8k | 7473 | 748 | 500 |
| Math1-3 | 3504 | 500 | 500 |
| ProntoQA | 2880 | 300 | 200 |
| Game of 24 | 2217 | 500 | 300 |

Table 3: Summary of the seed dataset sizes, validation and test set sizes. For all datasets we use 1k data points per iteration for finetuning.

### D.1 SEED DATASET SIZES

We summarize the seed dataset sizes for all datasets used in our experiments in Table 3. The seed dataset $D_0$, is used for scoring and selecting data points to get the selected data $\bar{D}_t$. The selected data is then put forward for prompt-based synthetic data generation (Section 4.2). We set the validation and test sets to be from the original seed datasets. We use the resulting synthetic datasets $\hat{D}_t$ for SFT, we generate a fixed sized synthetic dataset to enable fair comparison between selection methods and assess data efficiency (Section 5.1).

### D.2 SYNTHETIC DATA GENERATION PROMPTS

We provide the prompts used for prompt-based synthetic data generation (described in Section 4.2) below for all datasets used in our experiments:

- `GSM8k` see Appendix D.2.1.

- `Math1-3` see Appendix D.2.2.

- `ProntoQA` see Appendix D.2.3.

- `Game of 24` see Appendix D.2.4.

### D.2.1 GRADE SCHOOL MATHS

Below is the prompt we use for synthetic question generation for `GSM8k` using a `GPT-4o` teacher. In the prompt below $\{0\}$ are few-shot examples of questions and answers: $\{z_i\}_{i=1}^{k} \sim D_0$, we set $k = 5$ for all our experiments and $\{1\}$ is the question from the data selected by the student: $\bar{x} = \bar{z}[0]$ where $\bar{z} \sim \bar{D}_t$. The few-shot examples are formatted as follows: `#Given Instruction#: {} #Answer#: {}`

```
GSM8k synthetic question generation prompt

I want you to act as Instruction Creator.
Your objective is to rewrite a #Given Instruction# into a
more complex version, to make it a bit harder.
The #Rewritten Instruction# must be reasonable and must be
understood and responded to by humans.
Here are some #Examples#:
{0}
I want you to act as Instruction Creator.
Your objective is to rewrite a #Given Instruction# into a
more complex version, to make it a bit harder.
The #Rewritten Instruction# must be reasonable and must be
understood and responded to by humans.
You MUST complicate the #Given Instruction# using the
following method:
1.  Change the names of people #Given Instruction#.
2.  Change the objects in the #Given Instruction#.
3.  Change any quantities and durations in the #Given
Instruction#.
4.  Add 1 to 3 more operations in #Rewritten Instruction#.
5.  Change the operations, for example:  multiplication,
division, subtraction, addition, percentages, fractions and
combinations of these.
6.  You should try your best not to make the #Rewritten
Instruction# become verbose, #Rewritten Instruction# can only
add 10 to 20 words into #Given Instruction#.
Use #Examples# to complicate #Given Instruction#.
'#Given Instruction#', '#Rewritten Instruction#', 'given
instruction' and 'rewritten instruction' are not allowed to
appear in #Rewritten Instruction#.
#Given Instruction#:
{1}
#Rewritten Instruction#:
```

We use the following prompt to obtain synthetic answers from our `GPT-4o` teacher (and from our student model):

```
GSM8k answer prompt

Question:  {} Solve the problem step-by-step.  Answer:
```

### D.2.2  MATH1-3

Below is the prompt we use for synthetic question generation for `Math1-3` using a `GPT-4o` teacher, {0} are few shot examples of questions, answers and the type of problem e.g. Geometry, Algebra etc. The number of few-shot examples is 5 and are of the same type as the seed question. In the prompt below {1} is the type of mathematics problem and {2} is the question from the selected dataset: $\bar{x} = \bar{z}[0]$ where $\bar{z} \sim \bar{D}_t$. The few-shot examples are formatted as follows: `The type of math problem is {}.  #Given Instruction#:  {}` `#Answer#:  {}`

```
Math1-3 synthetic question generation prompt

I want you to act as an Instruction Creator for {1}
mathematics problems.
Create a new question #Rewritten Instruction# by using #Given
Instruction# as inspiration.  The new question should have a
single unique answer.
Ensure that the type of the question you generate #Rewritten
Instruction# matches the type of instruction #Given
Instruction#.
Make #Rewritten Instruction# different from #Given
Instruction#.
The #Rewritten Instruction# must be reasonable, have a
solution and must be understood and responded to by humans.
Here are some #Examples#:
{0}
Use #Examples# as inspiration to make #Rewritten Instruction#
different to #Given Instruction#.
'#Given Instruction#', '#Rewritten Instruction#', 'given
instruction' and 'rewritten instruction' are not allowed to
appear in #Rewritten Instruction#.
#Given Instruction# is a {1} math problem.
#Given Instruction#:
{2}
#Rewritten Instruction#:
```

We use the following prompt for obtaining synthetic answers from our `GPT-4o` teacher (and for obtaining answers from our student model):

```
Math1-3 answer prompt

Can you solve the following math problem?  {0}.  Provide a
bullet point summary of your reasoning.  Your final answer
should be a single answer, in the form \boxed{answer}, at the
end of your response.
```

### D.2.3 PRONTOQA

We present the prompt we use for synthetic question generation using a `GPT-4o` teacher for the `ProntoQA` dataset (Saparov & He, 2023). A datapoint from the `ProntoQA` dataset is comprised of a context, question and answer $z = (x = (c, q), y)$ where $x$ is comprised of the context $c$ and question $q$. The answers $y$ are boolean. The few-shot question generation is therefore comprised of contexts and questions for the teacher to generate new synthetic context and questions, $\hat{x}$. In the prompt below $\{0\}$ are few-shot examples of questions and answers from $\{z_i\}_{i=1}^{k} \sim D_0$, we set $k = 5$ for all our experiments and $\{1\}$ is the question from the selected dataset $\bar{x} = \bar{z}[0]$ where $\bar{z} \sim \bar{D}_t$. The few-shot examples $\{0\}$ are formatted as follows: `Context: {} Question: {}`.

```
ProntoQA synthetic question generation prompt

I want you to act as an Instruction Creator for logical
problems.
Create a new question #Rewritten Instruction# by using #Given
Instruction# as inspiration.
Make #Rewritten Instruction# different from #Given
Instruction# by changing the names, objects and adjectives.
Also vary the number of logical reasoning steps in #Rewritten
Instruction#.  Ensure that it is possible to answer the
question with true or false answer.
The #Rewritten Instruction# must be reasonable, have a
solution and must be understood and responded to by humans.
Here are some #Examples#:
{0}
Use #Examples# as inspiration to make #Rewritten Instruction#
different to #Given Instruction#.
'#Given Instruction#', '#Rewritten Instruction#', 'given
instruction' and 'rewritten instruction' are not allowed to
appear in #Rewritten Instruction#.
#Given Instruction#:
{1}
#Rewritten Instruction#:
```

We use the following prompt for obtaining synthetic answers from the `GPT-4o` teacher (and for obtaining answers from our student model):

```
ProntoQA answer prompt

Context:  {} Let's think step by step.  Response:
```

### D.2.4   GAME OF 24

Below is the prompt we use for synthetic question generation using `GPT-o3-mini` teacher for the `Game of 24` dataset. A datapoint from the `Game of 24` dataset is comprised of a set of four numbers and the arithmetic one-line solution to obtain 24. In the prompt below $\{0\}$ is the question: a set of numbers for instance $\bar{x} = [8, 8, 10, 12]$ and $\{1\}$ is the arithmetic answer for instance $\bar{y} = (12 - 10) \times 8 + 8$ where $\bar{z} = (\bar{x}, \bar{y})$ and $\bar{z} \sim \bar{D}_t$. We use backward reasoning to to obtain a new question and answer to the `Game of 24` (see the prompt below). We verify that the synthetic answer evaluates to 24 and that all the numbers from the synthetic question are also present in the synthetic answer. Since backward reasoning for synthetic data generation produces both the question and the answer, we then prompt our teacher, `GPT-4o` in a second step, with both the synthetic question and answer to get a synthetic reasoning trace without any verification of the reasoning steps to construct our synthetic dataset $\hat{D}_t$ (in the second prompt below).

> ### Game of 24 synthetic question generation prompt
>
> ```
> I want you to act as an instruction creator.  I want you to
> write a new problem to the game of 24.
> The numbers {0} need to be used to obtain the number 24.
> Use each number once, even if a number is repeated use it
> multiple times, with the arithmetic operations +, -, *, /
> to obtain 24.  Here is how the above numbers {0} are used to
> obtain 24:  {1}.
>
> I want you to create a new problem to the game of 24 using
> {1}.  Let's use a backward thinking method.  Take two of the
> distinct numbers in {1}.  Call them a and b.  Then construct
> an equation with two unknowns, a and b.  Pick integer values
> for the first variable b then solve for a.
>
> For example the numbers 8, 8, 10, 13 can be used to get
> 24:  13*8-10*8=24.  We can construct the following equation
> a*b-10*8=24 by substituting a=13 and b=8.  Rearranging we get
> a=104/b.  Let's pick an integer which divides into 104 for b:
> b=4 therefore a=26.
> We also could have picked b=2 and so a=62.  Therefore one
> possible answer to the game of 24 using this backward method
> is \boxed{4*26-10*8}.  If no answer is possible return
> \boxed{null}.
>
> Here is the current solution {1} again.  Enclose the new
> equation which results in 24 in \boxed{}.  Let's use this
> backward thinking method and think step by step.
> ```

Game of 24 prompt for synthetic reasoning steps

```
Use numbers and basic arithmetic operations (+ - * /) to
obtain 24.  Each step, you are only allowed to choose two
of the remaining numbers to obtain a new number.
Input:  4 4 6 8
Steps:
4 + 8 = 12 (left:  4 6 12)
6 - 4 = 2 (left:  2 12)
2 * 12 = 24 (left:  24)
Answer:  (6 - 4) * (4 + 8) = 24
Input:  2 9 10 12
Steps:
12 * 2 = 24 (left:  9 10 24)
10 - 9 = 1 (left:  1 24)
24 * 1 = 24 (left:  24)
Answer:  (12 * 2) * (10 - 9) = 24 Input:  4 9 10 13
Steps:
13 - 10 = 3 (left:  3 4 9)
9 - 3 = 6 (left:  4 6)
4 * 6 = 24 (left:  24)
Answer:  4 * (9 - (13 - 10)) = 24
Input:  1 4 8 8
Steps:
8 / 4 = 2 (left:  1 2 8)
1 + 2 = 3 (left:  3 8)
3 * 8 = 24 (left:  24)
Answer:  (1 + 8 / 4) * 8 = 24
Input:  5 5 5 9
Steps:
5 + 5 = 10 (left:  5 9 10)
10 + 5 = 15 (left:  9 15)
15 + 9 = 24 (left:  24)
Answer:  ((5 + 5) + 5) + 9 = 24
Input:  {question}
Here is the final answer:  {answer}
Provide the steps to obtain the final answer which equates
to 24, as if you did not have access to the answer.  Put your
final answer within \boxed{answer}.  Steps:
```

We use the following prompt to get answers from the student, similarly to Ni et al. (2025):

```
Game of 24 student prediction prompt

Use numbers and basic arithmetic operations (+ - * /) to
obtain 24.  Each step, you are only allowed to choose two
of the remaining numbers to obtain a new number.
Input:  4 4 6 8
Steps:
4 + 8 = 12 (left:  4 6 12)
6 - 4 = 2 (left:  2 12)
2 * 12 = 24 (left:  24)
Answer:  (6 - 4) * (4 + 8) = 24
Input:  2 9 10 12
Steps:
12 * 2 = 24 (left:  9 10 24)
10 - 9 = 1 (left:  1 24)
24 * 1 = 24 (left:  24)
Answer:  (12 * 2) * (10 - 9) = 24 Input:  4 9 10 13
Steps:
13 - 10 = 3 (left:  3 4 9)
9 - 3 = 6 (left:  4 6)
4 * 6 = 24 (left:  24)
Answer:  4 * (9 - (13 - 10)) = 24
Input:  1 4 8 8
Steps:
8 / 4 = 2 (left:  1 2 8)
1 + 2 = 3 (left:  3 8)
3 * 8 = 24 (left:  24)
Answer:  (1 + 8 / 4) * 8 = 24
Input:  5 5 5 9
Steps:
5 + 5 = 10 (left:  5 9 10)
10 + 5 = 15 (left:  9 15)
15 + 9 = 24 (left:  24)
Answer:  ((5 + 5) + 5) + 9 = 24
Input:  {question}
Put your final answer within \boxed{answer}.  Steps:
```

### D.3 EVALUATION PROMPTS

To assess whether the student's prediction is equal to the ground-truth answer we use gpt4o-mini to verify the correctness of the student. We use the following prompt and a system prompt which is different for each dataset used:

```
GSM8k and ProntoQA evaluation prompt

Question:{} Problem Setter's answer:{} Student's answer:{}
```

For GSM8k we use the following system prompt for evaluation, similarly to Mitra et al. (2024):

> **GSM8k evaluation system prompt**
>
> ```
> As an expert Math teacher, your role is to evaluate a
> student's answer to a word problem.  The problem is
> accompanied by a correct solution provided by the problem
> setter.  It is important to remember that there may be
> various methods to solve a word problem, so the student's
> steps might not always align with those in the problem
> setter's solution.  However, the final answer, typically
> a number, should be unique and match the problem setter's
> answer.  Your task involves analyzing the student's solution
> to identify any mistakes and determine whether the answer
> can be modified to correct the error.  If the student's
> answer is unfixable, consider creating practice problems
> to help improve their understanding.  Use the following
> format:  Error Analysis:  In one sentence, extract the final
> answer from the problem setter's solution and compare it
> with the student's answer.  Do they match?  Final Verdict:
> Correct/Incorrect.
> ```

For `ProntoQA` we use the following system prompt for evaluation:

> **ProntoQA evaluation system prompt**
>
> ```
> You are a logical expert.  Your role is to evaluate a
> student's answer to a logical reasoning problem.  The problem
> is accompanied by a correct solution provided by the problem
> setter.  Your task is to assess whether the problem setter's
> answer and the student's answer match.  Use the following
> format:  Error Analysis:  In one sentence, extract the final
> answer from the problem setter's solution and compare it
> with the student's answer.  Do they match?  Final Verdict:
> Correct/Incorrect.
> ```

If the output contains string variations of `"Final Verdict:  Correct"` then the student's prediction is correct and wrong otherwise. For the `Math1-3` and `Game of 24` datasets we use pattern matching to extract the student's answer and compare to the ground truth, see Section 5.1 for details.

