# OpenReview forum: "Towards Active Synthetic Data Generation for Finetuning Language Models"
_ICLR.cc/2026/Conference — Submitted to ICLR 2026_

### Official Review · Reviewer_3RBg · 2025-10-28

**Soundness:** 2
**Presentation:** 3
**Contribution:** 3
**Rating:** 4
**Confidence:** 4

**Summary:**

This paper studies an iterative approach to synthetic data generation for fine-tuning small language models (SLMs). Instead of generating a large static dataset from a teacher model at once, the authors propose a closed-loop scheme where the student model’s current state guides which examples are selected for further data generation by the teacher.
The work benchmarks several existing data selection strategies from active learning—uncertainty sampling (high-loss), reward-based scoring, LLM-as-a-judge, and BADGE diversity selection—and claims that simple heuristics like high-loss sampling outperform more expensive LLM-judge–based methods. Experiments are conducted on four reasoning datasets (GSM8K, Math1–3, ProntoQA, Game of 24) using various small instruction-tuned models. Results suggest that simple heuristics such as high-loss selection outperform more complex and expensive methods like LLM-as-a-judge, offering improved data efficiency under a fixed compute budget.

**Strengths:**

•	The authors provide a solid benchmark showing that simple, inexpensive heuristics (e.g., high-loss selection) can outperform more complex LLM-as-a-judge strategies.
•	The work provides practical guidance for synthetic data generation under constrained compute budgets, which can be valuable for practitioners training SLMs.
•	The paper is clearly written and easy to reproduce.
•	It contributes to the empirical understanding of how different data-selection heuristics impact fine-tuning performance and efficiency.

**Weaknesses:**

•	Limited novelty: The core idea—iterative, student-aware synthetic data generation—has been explored in multiple prior works. This paper mainly repackages it under the active learning perspective.
•	Lack of theoretical or conceptual insight: The paper does not explain why the compared heuristics differ or what properties (difficulty, diversity, informativeness) they capture.
•	Marginal performance gains: Improvements are small or inconsistent. For GSM8K and ProntoQA, performance remains below or comparable to prior SFT results; only one dataset (Game of 24) shows notable gains.
•	Scope limitation: All experiments are conducted on small 7–8B models; scalability to larger models or broader domains is untested.
•	Potential bias amplification: Since selection is based on student performance, the loop can reinforce sampling bias (e.g., favoring easy samples), which the paper neither analyzes nor mitigates.
•	Unclear takeaway: The results show minor absolute improvements, so the main claimed advantage, data efficiency, needs stronger quantitative justification.

**Questions:**

1.	How do you ensure that the iterative selection process does not bias the dataset, e.g., easier questions?
2.	Why does the performance of your proposed method have such a better performance, even better than the teacher model, on the Game of 24 dataset?
3.	How would the method behave with a larger teacher model?
4.	In your comparison, you do not control the SFT dataset size. Will that cause unfair comparison among different methods?

---

> ### Author Response · Authors · 2025-11-21
> **Author Response (1/n)**
>
> > Limited novelty: The core idea—iterative, student-aware synthetic data generation—has been explored in multiple prior works. This paper mainly repackages it under the active learning perspective.
>
> We appreciate the reviewer’s point. Iterative, student-aware synthetic data generation was introduced in prior work (e.g., LLM2LLM [1] and Lion [2]), and we do not claim this idea as novel. Our contribution is to provide the first unbiased and rigorous benchmark of selection strategies in this setting.
>
> Prior work does not disentangle the effect of data selection from synthetic data generation: LLM2LLM compares incorrect prioritization to using the ground-truth dataset, and Lion compares to a separate synthetic dataset (Vicuna), making it unclear whether gains are due to the selection strategy or the synthetic generation process itself. In contrast, we compare a range of selection methods directly against random sampling/static selection under identical training budgets, allowing performance improvements to be attributed to the selection strategy.
>
> Our findings show that (1) several selection methods significantly outperform random sampling, (2) simple criteria—such as high student loss—can outperform expensive LLM-as-a-judge scoring, and (3) properties of the student-selected data are retained in the synthetic data, demonstrating that the student can meaningfully steer generation quality (this new analysis is included in the updated manuscript (Section 5.4.3) see global response to all reviewers for further details).
>
> [1] Lee, Nicholas, et al. "Llm2llm: Boosting llms with novel iterative data enhancement." Findings of the Association for Computational Linguistics: ACL 2024. 2024.
>
> [2] Jiang, Yuxin, et al. "Lion: Adversarial distillation of proprietary large language models." arXiv preprint arXiv:2305.12870 (2023).
>
> > Lack of theoretical or conceptual insight: The paper does not explain why the compared heuristics differ or what properties (difficulty, diversity, informativeness) they capture.
>
> Thank you for the comment. To address this concern, we added two new analyses to explain why different heuristics behave differently.
>
> 1. We show that the synthetic data **retains key properties of the data selected by the student**, despite the noisy generation process (rewriting questions, simplifying/complicating questions, adding chain-of-thought for example). Thus, if a heuristic selects high-loss examples, the resulting synthetic data also has a high loss, and similarly for other scorers. As a result the scoring induces different biases on the synthetic data which in turn results in different performance.
> 2. We analyze why certain heuristics outperform others. The most effective methods are those that prioritize difficult data points. For example, high-loss and low-reward scoring generate **synthetic datasets on which the student achieves lower accuracy than random sampling**, indicating that the teacher produces harder data in response to these choices. Training on such challenging data enables the student to learn more and generalize better.
>
> This new analysis has been included in Section 5.4.3. These results show that selection algorithms that use properties like difficulty will in turn allow the teacher to generate data that is also difficult.
>
> > Marginal performance gains: Improvements are small or inconsistent. For GSM8K and ProntoQA, performance remains below or comparable to prior SFT results; only one dataset (Game of 24) shows notable gains.
>
> Thank you for the observation. Our focus is on data efficiency, not raw SFT performance. As defined in Section 2, meaningful efficiency claims can only be made when methods use (i) the same LLM backbone and (ii) comparable training budgets. Prior work on GSM8K achieves higher raw accuracy primarily because it uses hundreds of thousands to millions of training examples, whereas our iterative synthetic data generation operates up to 10k samples. Similarly, the ProntoQA comparison in Table 1 involved a different backbone making conclusions about efficiency unreliable (Qwen 1.5 has 46.4% zero-shot accuracy versus Qwen 2.5 with 90.6%, making comparison misleading).
>
> To avoid misleading comparisons, we have excluded baselines that use much larger datasets or different model backbones. Since some datasets have no prior work to compare against, we have removed discussion of state-of-the-art performance and moved Table 1 to the appendix (now Table 2 in the appendix in the updated manuscript). Under fair comparisons, iterative synthetic data generation is the most data-efficient approach on 2 of the 4 datasets as we are unaware of prior works in SFT with better performance using the same or less training data. Also see our global response to all reviewers for further discussion https://openreview.net/forum?id=U0I590wrsm&noteId=cZXssZhidf.

---

> ### Author Response · Authors · 2025-11-21
> **Author Response (2/n)**
>
> > Scope limitation: All experiments are conducted on small 7–8B models; scalability to larger models or broader domains is untested.
>
> We agree and acknowledge that our experiments are limited to 7–8B models due to compute constraints. Within these limits, we evaluate four datasets across four student LMs and report results over three runs to ensure statistically meaningful comparisons of data efficiency. Additionally our overall strategy—outlined in our introduction—is to obtain best results for SLMs for inference efficiency purposes, working with larger student models means these inference efficiencies will diminish.
>
> > Potential bias amplification: Since selection is based on student performance, the loop can reinforce sampling bias (e.g., favoring easy samples), which the paper neither analyzes nor mitigates.
>
> Thank you for raising this point. Conditioning selection on the student does indeed introduce a distribution shift or sampling bias, as discussed in prior work on covariate shift and active learning [1, 2]. In our setting, this bias is intentional: the goal is to steer the synthetic data toward examples that are most beneficial for the student rather than to preserve the original distribution. Empirically, this bias is desirable: the actively selected synthetic data yields substantially better data efficiency than random sampling, where no distribution shift occurs.
>
> [1] Sugiyama, Masashi, Matthias Krauledat, and Klaus-Robert Müller. "Covariate shift adaptation by importance weighted cross validation." Journal of Machine Learning Research 8.5 (2007).
>
> [2] Farquhar, Sebastian, Yarin Gal, and Tom Rainforth. "On statistical bias in active learning: How and when to fix it." ICLR (2021).
>
> > ... the loop can reinforce sampling bias (e.g., favoring easy samples) ...
>
> > How do you ensure that the iterative selection process does not bias the dataset, e.g., easier questions?
>
> We show in 5.4.4 in the original submission and Appendix C.7 in the updated manuscript that selection of hard samples outperforms selecting easy samples. Additionally, in our new analysis in Section 5.4.3 we show that selecting hard samples translates to generating hard synthetic data. Training on hard synthetic data then results in better performance. So selecting hard samples, not easy samples is desirable.
>
> > Unclear takeaway: The results show minor absolute improvements, so the main claimed advantage, data efficiency, needs stronger quantitative justification.
>
> Thank you for the comment. To provide a stronger quantitative justification of data efficiency, we report results over three independent runs and construct confidence intervals, which enables statistically reliable comparisons in the pairwise win-rate matrices. This methodology follows the protocol introduced in Ash et al. [1].
>
> In addition to accuracy differences in the learning curves and winrates (Figures 3 and 4), we now include a more intuitive metric: how much additional data random sampling requires to match the accuracy of the best active scorer. As shown in Figure 10 (now added to the appendix), **random sampling requires 33% to 100% more data than the top-performing selection method**, depending on the dataset. This highlights that even modest absolute differences in accuracy correspond to substantial gains in data efficiency.
>
> [1] Ash, Jordan T., et al. "Deep batch active learning by diverse, uncertain gradient lower bounds." ICLR (2020).
>
> > Why does the performance of your proposed method have such a better performance, even better than the teacher model, on the Game of 24 dataset?
>
> On Game of 24, the teacher’s accuracy is low, so we rely on a verifier during synthetic generation. Only generations that both evaluate to 24 and use all numbers in the prompt are accepted into the training set. This filtering allows the student to learn from verified-correct solutions, which can lead to performance exceeding that of the teacher itself.
>
> > How would the method behave with a larger teacher model?
>
> This is an interesting question, if we use a bigger teacher model we can expect the domain performance to improve by invoking scaling laws. As a result the teacher can generate synthetic datasets that will be more correct. In turn, finetuning the student on this synthetic data should strictly improve student performance as a result.

---

> > ### Author Response · Authors · 2025-11-21
> > **Author Response (3/n) n=3**
> >
> > > In your comparison, you do not control the SFT dataset size. Will that cause unfair comparison among different methods?
> >
> > Thank you for raising this point. Yes, we agree that fair data-efficiency comparisons require controlling the SFT dataset size. As defined in Section 2, we can only compare method $\alpha$ and method $\beta$ when (i) they are trained on the same number of examples, or (ii) method \beta achieves lower performance with more data, in which case $\alpha$ is more data-efficient. If $\beta$ achieves higher performance with more data, no data efficiency conclusion can be drawn without scaling $\alpha$ to comparable budgets.
> >
> > Table 1 in the original submission mixed baselines trained on different dataset sizes and different LLM backbones, which does not support a fair assessment of data efficiency. To avoid misleading claims we removed baselines that violate equal-budget or equal-backbone comparisons, as a result 2 datasets do not have comparisons with prior work and so we cannot make data efficiency claims with respect to prior related works. So we have de-emphasized discussion of state-of-the-art performance and moved Table 1 to Appendix C.2 (now Table 2). See the global reply to all reviewers for further discussion (https://openreview.net/forum?id=U0I590wrsm&noteId=cZXssZhidf).
> >
> > All of our results comparing static generation/random sampling and selection algorithms in Figures 3 and 4 (and throughout the main paper) compare methods at identical training budgets, ensuring fair evaluation. Just Table 1 didn’t control for SFT dataset sizes.

---

### Official Review · Reviewer_R7W1 · 2025-11-01

**Soundness:** 2
**Presentation:** 3
**Contribution:** 1
**Rating:** 2
**Confidence:** 5

**Summary:**

The paper studies how to make synthetic data generation for small language model (SLM) finetuning more data-efficient. Instead of static, one-shot generation from a teacher LLM, it proposes an iterative and student-aware approach: at each round, the student model is used to score or select informative examples from a seed set, which then guide new synthetic question–answer pairs generated by the teacher. The student is then finetuned on the accumulated synthetic data. The authors systematically compare several selection criteria—such as student loss, reward score, LLM-as-a-judge, and BADGE—and find that simple active learning strategies (e.g., selecting high-loss samples) outperform complex LLM-based scoring under the same compute budget. Experiments on four reasoning datasets (GSM8K, Math1–3, ProntoQA, and Game of 24) demonstrate strong data efficiency and even near-SOTA SFT performance with an order of magnitude less training data.

**Strengths:**

- The paper provides a clear and unified experimental framework for iterative synthetic data generation, grounding the idea in active learning principles.
- Empirical results are extensive: four datasets, four SLMs, multiple scoring algorithms, and comparisons to static generation.
- The conclusion—that simple, low-cost criteria (e.g., high student loss) outperform expensive LLM-as-a-judge scoring—is practical and well-supported.
- The ablation on selection design choices (argmax vs. sampling, using prediction vs. ground truth) is detailed and informative.
- The method achieves impressive data efficiency, matching or exceeding prior SFT performance with far fewer examples.

**Weaknesses:**

- Conceptually, the idea of iterative, student-guided data generation is not entirely new. Prior works such as [1,2,3] (especially 1) have explored similar active distillation loops where the student model guides data selection or teacher queries. However, the present paper does not cite or discuss these connections, nor does it clarify what is fundamentally new beyond applying classic active learning heuristics in this context.
- The method, while empirically solid, lacks deeper theoretical or conceptual insight into why the high-loss criterion works best—most explanations remain empirical.
- The study focuses solely on SFT; it would be valuable to test whether the same iterative synthetic data idea scales to RLHF or continual training.
- Some experiments (e.g., Math1–3) show relatively small margins over static generation, suggesting the benefit may vary with domain or data diversity.
- The paper does not analyze the diversity or potential overfitting of the generated datasets across iterations.

[1] ELAD: Explanation-Guided Large Language Models Active Distillation. In Findings of the Association for Computational Linguistics: ACL 2024, pages 4463–4475, Bangkok, Thailand. Association for Computational Linguistics.

[2] Evolving knowledge distillation with large language models and active learning. arXiv preprint arXiv:2403.06414.

[3] Active large language model-based knowledge distillation for session-based recommendation. In Proceedings of the AAAI Conference on Artificial Intelligence (Vol. 39, No. 11, pp. 11607-11615).

**Questions:**

- How does the proposed iterative generation compare quantitatively with recent active distillation frameworks (e.g., ELAD or Evolving KD) under similar budgets?
- Could the authors clarify whether the benefit primarily comes from better data selection or progressive curriculum effects from student feedback?
- How robust is the approach when the teacher’s generation quality degrades (e.g., smaller or domain-specific teachers)?
- Would the same active selection principles extend to reinforcement-based finetuning or continual pretraining?
- Could the authors release intermediate synthetic datasets to verify the claimed data-efficiency curves and reproducibility?

---

> ### Author Response · Authors · 2025-11-21
> **Author Response (1/n)**
>
> > Conceptually, the idea of iterative, student-guided data generation is not entirely new. Prior works such as [1,2,3] (especially 1) have explored similar active distillation loops where the student model guides data selection or teacher queries. However, the present paper does not cite or discuss these connections, nor does it clarify what is fundamentally new beyond applying classic active learning heuristics in this context.
>
> Thank you for pointing out these related works. We will cite and discuss them appropriately. The papers [1–3] study **active distillation**, where the student guides **answer** generation from the teacher and is trained on $(x, \hat{y})$ pairs. In contrast, our work focuses on **active synthetic data generation**, where the student guides **question–answer** generation and is trained on $(\hat{x}, \hat{y})$ pairs. This setting has received far less attention.
>
> We do not claim active synthetic data generation as a novel contribution—this was introduced in prior work [4, 5]. Our contribution is to provide the first **rigorous benchmark** comparing to canonical active learning strategies and existing approaches in this setting. We show that (i) iterative generation outperforms static generation (random sampling), (ii) simple criteria such as high student loss outperform expensive LLM-as-a-judge scoring, and (iii) synthetic data is steerable—the student can influence properties of the generated data (see new analysis introduced in the global response to all reviewers https://openreview.net/forum?id=U0I590wrsm&noteId=cZXssZhidf).
>
> [1] ELAD: Explanation-Guided Large Language Models Active Distillation. In Findings of the Association for Computational Linguistics: ACL 2024, pages 4463–4475, Bangkok, Thailand. Association for Computational Linguistics.
>
> [2] Evolving knowledge distillation with large language models and active learning. arXiv preprint arXiv:2403.06414.
>
> [3] Active large language model-based knowledge distillation for session-based recommendation. In Proceedings of the AAAI Conference on Artificial Intelligence (Vol. 39, No. 11, pp. 11607-11615).
>
> [4] Lee, Nicholas, et al. "Llm2llm: Boosting llms with novel iterative data enhancement." Findings of the Association for Computational Linguistics: ACL 2024. 2024.
>
> [5] Jiang, Yuxin, et al. "Lion: Adversarial distillation of proprietary large language models." arXiv preprint arXiv:2305.12870 (2023).
>
> > The method, while empirically solid, lacks deeper theoretical or conceptual insight into why the high-loss criterion works best—most explanations remain empirical.
>
> Thank you for the suggestion, we have added new conceptual insights to the manuscript which we highlight in the global response to all reviewers (https://openreview.net/forum?id=U0I590wrsm&noteId=cZXssZhidf) and add this new analysis to the manuscript in Section 5.4.3. To summarize:
> * We show that the synthetic data retains key properties of the data selected by the student on average, despite the noisy generation process (rewriting questions, simplifying/complicating questions, adding chain-of-thought etc.). Thus, if an active method selects high-loss examples, the resulting synthetic data also has a high loss and reflects this difficulty profile, and similarly for other scorers such as using a reward. As a result the scoring induces different biases on the synthetic data which results in different performance.
> * We analyze why certain active learning heuristics outperform others. The most effective methods are those that prioritize difficult data points. For example, high-loss and low-reward scoring generate synthetic datasets on which the student achieves lower accuracy than random sampling, indicating that the teacher produces harder data in response to the selected data. Training on such challenging data enables the student to learn more and generalize better.
>
> Both of these insights explain why selecting data prior to synthetic data generation is a good idea. Synthetic data that has similar properties to our selected data and it is harder for the student, therefore the synthetic data enhances student performance upon finetuning.
>
> > The study focuses solely on SFT; it would be valuable to test whether the same iterative synthetic data idea scales to RLHF or continual training.
>
> Thank you for the suggestion, we agree that extending iterative synthetic data generation to RLHF or continual training would be a valuable direction. There are clear parallels to exploration strategies in RL that merit future investigation. However, this is beyond the scope of the present work, which focuses on SFT. We have noted this explicitly in the limitations section (Section 6).

---

> ### Author Response · Authors · 2025-11-21
> **Author Response 2/n**
>
> > Some experiments (e.g., Math1–3) show relatively small margins over static generation, suggesting the benefit may vary with domain or data diversity.
>
> Thank you for the observation. We agree that the benefits of active selection can be domain-dependent, as is well established in traditional active learning. In our study, iterative synthetic generation yields larger gains on GSM8k than on Math1–3, indicating that some domains are more amenable to active strategies than others. Moreover, different domains favour different selection criteria; in the settings we evaluate, prioritizing difficult samples (high uncertainty/high loss) performs best overall.
>
> We have added an additional analysis in the paper in Appendix C.3 where we measure the percentage increase in the amount of data static selection/random sampling requires to obtain the same performance as certain active selection methods; we show that static generation requires 40% more data to obtain the same performance as high uncertainty sampling for Math1-3 which is a substantial increase in the amount of data required.
>
> > The paper does not analyze the diversity or potential overfitting of the generated datasets across iterations.
>
> We do not observe evidence of overfitting to the synthetic datasets as validation and test accuracy both improve monotonically across iterations (Figure 3), suggesting that performance gains are not driven by overspecialization to the generated data. It is worth mentioning that when the student produces generations and then trains on them there is overfitting [1], since we use a separate teacher this is not observed in our work.
>
> [1] Shumailov, Ilia, et al. "AI models collapse when trained on recursively generated data." Nature 631.8022 (2024): 755-759.
>
> > How does the proposed iterative generation compare quantitatively with recent active distillation frameworks (e.g., ELAD or Evolving KD) under similar budgets?
>
> Thank you for the suggestion. We will include a comparison with Evolving KD and expect to have results ready before the end of the rebuttal period. We will let you know when the results have been included into the updated manuscript. Although the question refers to either ELAD or Evolving KD, we plan to evaluate both within our iterative synthetic data generation framework. ELAD requires a more involved implementation, so those results will be added in the camera-ready version.
>
> > Could the authors clarify whether the benefit primarily comes from better data selection or progressive curriculum effects from student feedback?
>
> Thank you for the question. In our framework, both effects are intertwined: the student guides data selection, the selected examples drive synthetic data generation, and the updated student is then used for selection in the next iteration. This naturally creates a progressive curriculum.
>
> Our results indicate that the primary benefit comes from **better data selection**: there are active methods that outperform static generation which has no student feedback or curriculum. Also, as shown in Figure 7, the most effective methods are those that select examples leading to **more difficult synthetic data**, from which the student learns more during fine-tuning. The progressive curriculum emerges as a consequence of improving selection quality over iterations.
>
> > How robust is the approach when the teacher’s generation quality degrades (e.g., smaller or domain-specific teachers)?
>
> Our approach assumes access to a high-quality teacher, as student performance is fundamentally bounded by the quality of the synthetic data it receives (in the absence of a verifier). When the teacher’s generation quality degrades e.g., using a smaller or domain-specialized model, the quality of the synthetic data will also degrade, and the student will ultimately inherit this limitation.
>
> The motivation of our setting is precisely to transfer the strengths of a powerful teacher to a smaller student model. Using a low-quality teacher would not achieve this goal and would yield limited benefit. This dependency is listed in our limitations section, Section 6.

---

> > ### Author Response · Authors · 2025-11-21
> > **Author Response (3/n) n=3**
> >
> > > Would the same active selection principles extend to reinforcement-based finetuning or continual pretraining?
> >
> > Thank you, great question! Extending active selection principles to reinforcement-based fine-tuning or continual pretraining is an interesting direction, and we see clear conceptual parallels. For example, between active selection and exploration strategies in RL. However, these settings are beyond the scope of the present paper, which focuses specifically on SFT with synthetic question–answer generation.
> >
> > For continual pretraining or document-level generation, a practitioner could analogously treat the first part of a document as the question and generate the continuation $\hat{y}$​, applying active selection to prioritize which seed documents or segments to train on directly or to synthetically generate. We agree this is a promising avenue for future work and have mentioned this in Section 6.
> >
> > > Could the authors release intermediate synthetic datasets to verify the claimed data-efficiency curves and reproducibility?
> >
> > Yes, we will release our code as well.

---

> > > ### Author Response · Authors · 2025-11-28
> > > **Evolving Knowledge Distillation Results**
> > >
> > > > How does the proposed iterative generation compare quantitatively with recent active distillation frameworks (e.g., ELAD or Evolving KD) under similar budgets?
> > >
> > > We have added Evolving KD as a comparison to the other active synthetic data generations methods in Appendix C.1 of the updated paper. We have also highlighted this new result to all reviewers in the following global comment: https://openreview.net/forum?id=U0I590wrsm&noteId=aTeYe7LvDA.
> > >
> > > Although the original query refers to comparing either ELAD or Evolving KD, we plan to evaluate both within our iterative synthetic data generation framework. ELAD will be added in the camera-ready version.

---

### Official Review · Reviewer_hUid · 2025-11-04

**Soundness:** 2
**Presentation:** 3
**Contribution:** 2
**Rating:** 4
**Confidence:** 4

**Summary:**

This paper introduces an iterative, closed-loop method for Active Synthetic Data Generation (ASDG) to finetune Small Language Models (SLMs) using a Teacher LLM. The process leverages the Student's current performance (loss and predictions) to actively select seed prompts for generating new, highly informative synthetic data. The authors formally benchmark various data selection heuristics (including loss, reward, LLM-as-a-judge, and BADGE) across four reasoning datasets and four SLMs. The main finding is that iterative generation is significantly more data-efficient than static generation, and the simple heuristic of prioritizing samples with the highest student loss is the most effective and performant, often achieving competitive results with much larger, statically generated SFT datasets.

**Strengths:**

The conclusion that simple data selection methods, such as prioritizing hard samples with high loss, often outperform complicated and expensive LLM-as-a-judge based methods is a useful result for practitioners, suggesting that resource-intensive scoring is not always necessary.

The use of learning curves and the pairwise win-rate matrix (Figure 4) provides a structured comparative analysis focused on the core concept of data efficiency.

Analysis is thorough, covering four distinct reasoning datasets (GSM8k, Math1-3, ProntoQA, Game of 24) and four different student SLMs (Mistral 7B, Llama 3 8B, Qwen 7B, Qwen 2.5 7B).

**Weaknesses:**

The paper claims to provide a "benchmark study for iterative synthetic data generation" but fails to run a head-to-head comparison against the actual selection methods proposed by the most relevant prior works, specifically LLM2LLM (Lee et al., 2024) and the full LION (Jiang et al., 2023c) strategy. The critical "incorrect student answers" criterion from LLM2LLM is relegated to the appendix (C.1) despite being a highly competitive baseline in a truly active synthetic data setting.

Limitations in Experimental Scale and Baseline Selection Validity: 1) The paper focuses on a small, fixed training budget (1k samples per iteration, max total 10k for GSM8k/Math1-3). This scale is extremely small for finetuning modern SLMs, especially when compared to the multi-hundred thousand to multi-million sample sizes used by SOTA baselines in Table 1. While the relative performance of the selection methods within this small budget is clear, can the absolute efficiency claims scaling up with data? 2) Static Generation (Random Sampling) is Too Weak: This is the weakest possible baseline. A more competitive baseline would be a fixed synthetic dataset filtered using some non-active metric (e.g., high reward score, or high diversity selection applied once). The superiority of any student-aware curriculum over pure random sampling is expected, so the magnitude of this win is not fully persuasive.

Insufficient Discussion of Computational Budget and Trade-offs The paper frames the work around a "fixed data generation budget" (L14). However, the budget analysis is incomplete. 1) Teacher Output Cost is Missing: The current analysis (Figure 5) only uses Input Tokens as a proxy for compute. Since the primary cost for Chain-of-Thought (CoT) generation is the Teacher's Output Tokens, omitting this cost makes the true teacher compute comparison incomplete. 2) Student Cost is Ignored: The iterative loop requires the student to make predictions and compute gradients on the entire seed set $D_{0}$ at every iteration. This computational overhead on the student's side grows with the seed set size and the number of iterations ($|D_0| \times T$), and should be explicitly discussed in the efficiency trade-off analysis.

**Questions:**

The authors state they have access to the ground-truth label $y$ but find that using the loss from the model's own generation $\mathcal{L}(z_i, \theta)$ ("uncertainty") is more effective than the true loss $\mathcal{L}(z_i, y)$. This counter-intuitive finding is not explored beyond a single sentence. Why is using the model's own (potentially incorrect) generation as the target label better than using the verified ground truth label? This requires deeper analysis.

Why BADGE Implementation? The critical decision to use generated sequences instead of ground-truth targets for BADGE's gradient representations (L243-245) lacks sufficient justification, similar to the loss/uncertainty choice.

---

> ### Author Response · Authors · 2025-11-21
> **Author Response (1/n)**
>
> > The paper claims to provide a "benchmark study for iterative synthetic data generation" but fails to run a head-to-head comparison against the actual selection methods proposed by the most relevant prior works, specifically LLM2LLM (Lee et al., 2024) and the full LION (Jiang et al., 2023c) strategy. The critical "incorrect student answers" criterion from LLM2LLM is relegated to the appendix (C.1) despite being a highly competitive baseline in a truly active synthetic data setting.
>
> In our manuscript we compare to both LLM2LLM [1] (we call this “incorrect” prioritization in Appendix C.1) and to full Lion [2], referred to as Lion in Figures 3 and 4. We run experiments multiple times and produce confidence intervals to run rigorous head-to-head comparisons with both of these methods in Figure 4 and Figure 8 (with LLM2LLM). We decided to compare LLM2LLM's incorrect student response scoring in the appendix since it uses ground truth labels for scoring unlike the other active selection methods like Lion that do not require labels. LLM2LLM’s incorrect prioritisation is indeed highly competitive and obtains good results in 3/4 datasets however obtains poor performance on GSM and so obtains poor results in head-to-head winrates (Figure 8).
>
> [1] Lee, Nicholas, et al. "Llm2llm: Boosting llms with novel iterative data enhancement." Findings of the Association for Computational Linguistics: ACL 2024. 2024.
>
> [2] Jiang, Yuxin, et al. "Lion: Adversarial distillation of proprietary large language models." arXiv preprint arXiv:2305.12870 (2023).
>
> > 1) The paper focuses on a small, fixed training budget (1k samples per iteration, max total 10k for GSM8k/Math1-3). This scale is extremely small for finetuning modern SLMs, especially when compared to the multi-hundred thousand to multi-million sample sizes used by SOTA baselines in Table 1. While the relative performance of the selection methods within this small budget is clear, can the absolute efficiency claims scaling up with data?
>
> We appreciate the reviewer’s point. Our goal is not to train state-of-the-art SLMs with million-sample budgets, but to provide a controlled and unbiased benchmark of data efficiency for iterative synthetic data generation. The focus of the paper is to determine whether active synthetic generation outperforms static selection (random sampling) and to offer practical guidance to practitioners operating under realistic compute constraints. Prior work does not isolate this question: Lion compares against Vicuna, and LLM2LLM compares against the original seed dataset, so neither disentangles data selection from synthetic generation as we do.
>
> We agree—also in line with Reviewer 3RBg—that raw performance comparisons against methods using vastly larger datasets are not meaningful. Data-efficiency claims can only be made when methods use comparable training sizes; if method \alpha underperforms method \beta despite using more data, conclusions can be drawn (\beta is more data efficient), but the reverse is not possible without scaling budgets. Accordingly, we have removed discussion of state-of-the-art performance and excluded comparisons involving much larger datasets as data efficiency comparisons are not possible and data efficiency is the main goal of the paper. Notably, for 2 of the 4 datasets in Table 2, we are not aware of prior SFT approaches that use less data and achieve higher performance with the same LLM as the iterative synthetic data generation techniques we consider.
>
> That said, as outlined in our global response, we have increased the training budgets for Math1-3 from 10k to 14k and Game of 24 from 6k to 10k points.
>
> > While the relative performance of the selection methods within this small budget is clear, can the absolute efficiency claims scaling up with data?
>
> As we scale with larger and larger dataset sizes we expect performance gains to increase but the increases to get smaller and smaller.

---

> > ### Author Response · Authors · 2025-11-21
> > **Author Response (2/n) n =2**
> >
> > > 2) Static Generation (Random Sampling) is Too Weak: This is the weakest possible baseline. A more competitive baseline would be a fixed synthetic dataset filtered using some non-active metric (e.g., high reward score, or high diversity selection applied once). The superiority of any student-aware curriculum over pure random sampling is expected, so the magnitude of this win is not fully persuasive.
> >
> > We respectfully disagree that random sampling (RS) is an uninformative or uncompetitive baseline. RS corresponds to diversity-based selection that is not conditioned on the student, and in the active learning literature it is often a strong baseline e.g., in the original BADGE paper, RS outperforms roughly half of the active learning strategies evaluated (Fig. 4) [1]. In domains where all samples are similarly informative, RS is in fact optimal.
> >
> > RS is also the most relevant comparison for our setting: it is equivalent to statically generating a synthetic dataset once and then fine-tuning, so evaluating against RS is necessary to justify whether practitioners should prefer iterative synthetic data generation. It is also competitive with many of the methods we compare against e.g. Lion, (Figure 4).
> >
> > We also note that suggested alternatives such as high-reward or high-diversity filtering are typically conditioned on the student, making them active rather than purely static. For example, a high diversity is also active e.g. Lion is a type of diversity selection over LLM-as-a-judge scores or coresets that require student embeddings [2]. Finally, our benchmark treats all selection methods as baselines—including RS—and pits them against each other through pairwise win-rate matrices, ensuring fairness and transparency across baselines.
> >
> > [1] Ash, Jordan T., et al. "Deep batch active learning by diverse, uncertain gradient lower bounds." ICLR (2020).
> >
> > [2] Sener, Ozan, and Silvio Savarese. "Active learning for convolutional neural networks: A core-set approach." ICLR (2018).
> >
> > > 1) Teacher Output Cost is Missing: The current analysis (Figure 5) only uses Input Tokens as a proxy for compute. Since the primary cost for Chain-of-Thought (CoT) generation is the Teacher's Output Tokens, omitting this cost makes the true teacher compute comparison incomplete.
> >
> > Thank you for the suggestion, including both input and output tokens is indeed more thorough. We will run this and update Figure 5 to have input and output tokens. We will let you know when this is ready and the Figure has been updated.
> >
> > > 2) Student Cost is Ignored: The iterative loop requires the student to make predictions and compute gradients on the entire seed set  at every iteration. This computational overhead on the student's side grows with the seed set size and the number of iterations (), and should be explicitly discussed in the efficiency trade-off analysis.
> >
> > Thank you for raising this point. Our analysis follows a common assumption in active learning: the cost of data acquisition (here, synthetic data generation) dominates the cost of student evaluation. This assumption is reasonable in our setting, as the teacher model (e.g., GPT-4o, ~1T parameters) is orders of magnitude larger than the student SLMs. Nonetheless, we agree that the student-side computational overhead grows with the seed set and number of iterations, and we will clarify this assumption and its implications in the efficiency trade-off discussion (line 412).
> >
> > > The authors state they have access to the ground-truth label  but find that using the loss from the model's own generation  ("uncertainty") is more effective than the true loss . This counter-intuitive finding is not explored beyond a single sentence. Why is using the model's own (potentially incorrect) generation as the target label better than using the verified ground truth label? This requires deeper analysis.
> >
> > Thank you for the comment. We provided this analysis in Section 5.4.4 of the original manuscript (now Appendix C.7 in the updated version). As shown in Figure 14, using ground-truth labels (“gt”) yields no improvement—and in some cases underperforms—compared to using the student’s own predictions.
> >
> > >Why BADGE Implementation? The critical decision to use generated sequences instead of ground-truth targets for BADGE's gradient representations (L243-245) lacks sufficient justification, similar to the loss/uncertainty choice.
> >
> > The original BADGE implementation uses the generated sequences—it is an active learning algorithm. So to keep true to the original BADGE algorithm we also use generated sequences instead of the ground-truth targets. However, we did try this for certain datasets and didn’t get a significant improvement over using the original BADGE implementation, see Appendix C.7 for learning curves for GSM8k, ProntoQA and Game of 24.

---

> > > ### Author Response · Authors · 2025-11-28
> > > **Teacher Output Cost**
> > >
> > > > 1) Teacher Output Cost is Missing: The current analysis (Figure 5) only uses Input Tokens as a proxy for compute. Since the primary cost for Chain-of-Thought (CoT) generation is the Teacher's Output Tokens, omitting this cost makes the true teacher compute comparison incomplete.
> > >
> > > We have updated Figure 6 (Figure 5 in the original submission) to include both teacher input and output tokens as a proxy for the teacher compute cost for different active selection algorithms. The overall message remains the same: methods that rely on the teacher for scoring are far more expensive than active methods that do not.

---

### Author Response · Authors · 2025-11-21
**Global Response to Reviewers (1/2)**

We express our gratitude to the Chairs and the Reviewers for spending time reviewing our paper and providing constructive feedback. We are grateful to the Reviewers for recognizing firstly the importance of (1) the practical conclusion that **simple, inexpensive selection heuristics—particularly high-loss prioritization—often outperform more complex and resource-intensive LLM-as-a-judge scoring methods (hUid, R7W1, 3RBg)**, (2) the **clarity and rigor of the experimental framework**, including learning curves, pairwise win-rate matrices, and a unified setup for iterative synthetic data generation (hUid, R7W1), (3) the **breadth and depth of empirical evaluation**, spanning four reasoning datasets, four SLMs, multiple scoring algorithms, multiple seeds and extensive comparisons and ablations (hUid, R7W1), and (4) the **practical relevance of the guidance provided for data-efficient training**, especially under compute-constrained settings, which supports practitioners (R7W1, 3RBg). We are also grateful for constructive critical comments, which helped us to improve the paper!

# Active Synthetic Data Generation (R7W1, 3RBg)

We agree with the reviewers R7W1 and 3RBg that iterative, student-aware synthetic data generation of questions and answers ($\hat{z} = (\hat{x}, \hat{y})$) for SFT is not a novel contribution of our work. We do not claim to introduce these ideas either and we attribute it to prior work (LLM2LLM [1] and Lion [2]). Our contribution is to provide the first unbiased and rigorous benchmark of selection strategies in this setting, where we draw upon the active learning literature.

Prior work does not disentangle the effect of data selection from synthetic data generation: LLM2LLM compares incorrect prioritization to using the ground-truth dataset, and Lion compares to a separate synthetic dataset (Vicuna), making it unclear whether gains are due to the selection strategy or the synthetic generation process. In contrast, we compare a range of selection methods directly against random sampling/static selection under identical training budgets, allowing performance improvements to be attributed to a selection strategy.

[1] Lee, Nicholas, et al. "Llm2llm: Boosting llms with novel iterative data enhancement." Findings of the Association for Computational Linguistics: ACL 2024. 2024.

[2] Jiang, Yuxin, et al. "Lion: Adversarial distillation of proprietary large language models." arXiv preprint arXiv:2305.12870 (2023).

# Paper updates

All paper updates are highlighted in orange in the updated manuscript on Openreview.

## Additional conceptual insights (R7W1, 3RBg)

We have added two new analyses in Section 5.4.3 to better understand the dynamics of synthetic data generation:
1. First, we show that the synthetic data produced by the teacher statistically resembles the data selected by the student: although individual samples may be rephrased, simplified, or made more complex, with added chain-of-thought, there is a strong correlation at the dataset level, but only a weak correlation per data point. This indicates that synthetic generation preserves global scorer selection biases: if the selected data has a high loss then the synthetic data also has a high loss.
2. Second, we study the effects on the student and we observe that when the student prioritizes “difficult” data, the teacher also generates data that is difficult for the student. Specifically, prioritizing high-loss or low-reward samples yields synthetic data that produces lower accuracies than random sampling, demonstrating that the teacher generates harder data that ultimately enables greater learning and improved generalization.

Both of these observations explain why selecting data prior to synthetic data generation results in data that has similar properties to our selected data and therefore leads to enhanced student capabilities upon finetuning. Therefore this is more efficient than pruning the synthetic dataset after synthetic data generation.

## Significant gains in data efficiency (3RBg, R7W1)

Rather than comparing percentage accuracy differences in the learning curves, we measure the percentage difference in the amount of student data random sampling requires for equal performance with the best active synthetic data generation methods along the x-axis. This more clearly highlights the substantial performance gains from active selection; random selection requires from 33% to 100% more data than the best active selection methods. These results are now included in ~~Appendix C.3~~ Figure 3.

---

> ### Author Response · Authors · 2025-11-21
> **Global Response to Reviewers (2/2)**
>
> ## Unfair comparisons with prior work (hUid, 3RBg)
>
> We agree with reviewers hUid, 3RBg that raw performance comparisons against methods using vastly larger datasets are not meaningful. Data-efficiency claims can only be made when methods use comparable training sizes; if method $\alpha$ underperforms method $\beta$ despite using more data than $\beta$, we can say that $\beta$ is more data efficient. But if $\beta$ uses more data then it is not possible to draw data efficiency conclusions without scaling training budgets for $\alpha$ (as outlined in Section 2). Accordingly, we have removed comparisons for prior work which finetune on larger datasets as they do not enable a fair evaluation of data efficiency. We also remove comparisons involving different LLM backbones (e.g., the ProntoQA comparison, where we used Qwen 1.5 with 46.4% zero-shot accuracy versus a method using Qwen 2.5 with 90.6%, makes comparison misleading).
>
> For two of the four datasets, iterative synthetic data generation is the most data efficient method compared to prior work. We are not aware of prior work that uses less SFT data and achieves better performance than ours with the same LLM. However for the other 2 datasets we have no comparisons to prior work in SFT and so discussion about state-of-the-art performance is weaker now. As a result we have de-emphasized this claim in the paper. So we exclude comparisons that rely on larger dataset sizes than ours and different LLM backbones, as they do not enable a fair evaluation of data efficiency (to make space in the main paper for new conceptual analysis we have moved the table to Appendix C.2).
> ## Increased training budgets (hUid)
>
> We have increased the training budgets to allow for larger dataset sizes in iterative synthetic data generation: from 10k to 14k for Math1–3 and from 6k to 10k for Game of 24. We also increased the training budget for certain methods on GSM8k from 5k to 10k, so that they all train on up to 10k samples. Accordingly, we have updated the pairwise win-rate matrices in Figures 4 (and 8). High-loss selection continues to perform best, and all conclusions remain unchanged.
>
> ## BADGE with ground truth answers (hUid)
>
> We evaluated BADGE using ground-truth answers and found little benefit compared to using student predictions; the results are provided in Appendix C.7. Moreover, using student predictions is consistent with the intent of the original algorithm. For these reasons, our implementation relies on student predictions to obtain gradients.

---

> > ### Author Response · Authors · 2025-11-28
> > **Summary of New Requested Experiments**
> >
> > ## Evolving knowledge distillation (R7W1)
> >
> > We have added a comparison to **Evolving Knowledge Distillation (EvoKD)** [1], in which correct and incorrect student answers are sampled evenly for synthetic data generation, and we include this comparison in Appendix C.1. This method is similar to **LLM2LLM**, which prioritizes only incorrect samples [2]. We find that EvoKD performs worse than LLM2LLM on Math1–3 and ProntoQA and obtains similar performance to LLM2LLM on GSM and Game of 24. On the GSM dataset EvoKD performs on par with LLM2LLM and exhibits the same pathologies as they both underperform random sampling. Considering pairwise winrates (Figure 9), EvoKD underperforms methods that explicitly aim to promote difficult samples for synthetic data generation, such as high-loss and hard LLM-as-a-judge scoring.
> >
> > [1] Evolving knowledge distillation with large language models and active learning. arXiv preprint arXiv:2403.06414.
> >
> > [2] Lee, Nicholas, et al. "Llm2llm: Boosting llms with novel iterative data enhancement." Findings of the Association for Computational Linguistics: ACL 2024. 2024.
> >
> > ## Teacher output cost (hUid)
> >
> > Previously, we only considered the number of input tokens to the teacher as a proxy for the cost of calling the teacher. As was correctly pointed out, we did not account for the output cost. We have updated Figure 6 to include the sum of teacher input and output tokens as a proxy for the cost of using the teacher for each selection algorithm. The overall message remains the same: methods that rely on the teacher for scoring are far more expensive than active methods that do not.

---

### Author Response · Authors · 2025-11-28
**Discussion Conclusion**

**Dear Reviewers**,

We sincerely appreciate the time and effort you invested in evaluating our work. We carefully considered all feedback and have incorporated your suggestions to improve the clarity, rigor, and completeness of our submission. We believe these changes have strengthened the paper substantially.

Thank you again for your constructive comments.

**Dear Area Chairs**,

We believe we have fully addressed all reviewer concerns. All modifications to the manuscript are highlighted in orange for ease of reference, and our responses clarify each point raised during the review process.

Sincerely,
The Authors of Paper 13622

---

### Meta-Review · Area_Chair_bt7f · 2026-01-07

**Summary:**

This paper presents an iterative, student-aware framework for active synthetic data generation to fine-tune small language models and provides a careful empirical comparison of data selection heuristics.

Reviewers appreciated the clear presentation, reproducibility, and the practical finding that simple high-loss selection often outperforms costly LLM-as-a-judge methods. However, concerns center on limited novelty relative to prior active distillation work, missing or weakly positioned baselines, small experimental scale, and incomplete compute-budget analysis. These issues substantially weaken the paper’s claimed contributions and impact.

**Reviewer Concerns:**

Reviewers have concerns about limited novelty relative to prior active distillation work, missing baselines, and small experimental scale.

**Reviewer Scores:**

Reviewers generally keep their scores after rebuttal.

---

### Decision · Program_Chairs · 2026-01-26

Reject